# Inhibitory Effect of Fucoidan Analogs on Highly Metastatic Gastric Cancer Cells via Galectin-4 Inhibition

**DOI:** 10.3390/ijms26189228

**Published:** 2025-09-21

**Authors:** Shuting Ji, Maniyamma Aswathy, Yuya Kuboki, Yoshio Takada, Kazunobu Toshima, Daisuke Takahashi, Hiroko Ideo

**Affiliations:** 1Laboratory of Glycobiology, The Noguchi Institute, 1-9-7, Kaga, Itabashi, Tokyo 173-0003, Japan; jishuting@noguchi.or.jp (S.J.);; 2Department of Applied Chemistry, Faculty of Science and Technology, Keio University, 3-14-1, Hiyoshi, Kouhoku-ku, Yokohama 223-8522, Kanagawa, Japan; aswathymadhukrishnan@keio.jp (M.A.); dtak@applc.keio.ac.jp (D.T.)

**Keywords:** fucoidan, galectin-4, gastric cancer, molecular docking, peritoneal metastasis

## Abstract

In malignant-type gastric cancer, peritoneal dissemination is the most frequent metastatic process and is an inoperable condition for which effective treatment is lacking. Our research has revealed that galectin-4 plays an important role in the peritoneal metastasis of gastric cancer cells. Based on this, we hypothesized that inhibiting galectin-4 could suppress peritoneal metastasis. The inhibitory activity towards galectin-4 binding was evaluated using an enzyme-linked immunosorbent assay, while the suppressive effect on gastric cancer cell proliferation was assessed using an adenosine triphosphate-based cell viability assay. Direct binding to galectin-4 was examined by surface plasmon resonance analysis. Chemically synthesized fucoidan analogs exhibited significant suppressive activity against the proliferation of gastric cancer cells, partly via a galectin-4-mediated pathway. Among the 13 fucoidan analogs tested, analog **10**, whose sugar chains composed of repeating 2,3-*O*-sulfated α(1,4)-linked L-fucose, showed significant inhibitory activity against galectin-4 binding and cell proliferation. **14**, the cholestanol-conjugated analog **10**, exhibited a pronounced increase in inhibitory activity, consistent with potential multimerization. Molecular docking and site-directed mutagenesis studies revealed that Arginine-45 in galectin-4 is important for binding to fucoidan analogs. In conclusion, fucoidan analogs with a strong affinity for galectin-4 are promising candidates for inhibiting the peritoneal metastasis of galectin-4-positive gastric cancer cells.

## 1. Introduction

In malignant-type gastric cancer, peritoneal dissemination is a most frequent metastatic process within the peritoneal cavity. Peritoneal dissemination is a major cause of increased mortality; however, it is difficult to treat surgically, and an effective treatment has not yet been established. New molecular-targeted therapies are becoming available for gastric cancer, but are not effective for all patients [1].

Our previous study showed that galectin-4 plays an important role in the peritoneal dissemination of poorly differentiated gastric cancer cells [2]. Galectin-4 is mainly expressed in intestinal epithelial cells and has two carbohydrate-recognition domains (CRDs) with different binding specificities [3]. This allows galectin-4 to cross-link and regulate important molecules involved in various biological processes. The role of galectin-4 in stabilizing lipid rafts, transporting glycoproteins, and tumor progression has been studied [3,4]. Some reports have also suggested that galectin-4 expression is correlated with the malignant transformation of cancers in these tissues [5,6].

The suppression of peritoneal metastasis of gastric cancer was observed by the knockdown of galectin-4 expression in a mouse model [2]. Therefore, we hypothesized that inhibiting the binding activity of galectin-4 could also suppress the metastasis of gastric cancer. Our previous study showed that sulfation and fucosylation of carbohydrates enhanced the inhibitory activity toward galectin-4, suggesting that sulfated and fucosylated glycans may be candidates for galectin-4 inhibitors [7].

Fucoidan is a macromolecular polysaccharide widely found in brown algae that comprise sulfated fucose residues and have several biological functions [8,9,10]. The anticancer properties of natural fucoidans have been reported in both in vivo and in vitro studies for various types of cancers [11,12]. However, the studies on the pharmacodynamics of fucoidan remain relatively limited [13]. Fucoidan from *Fucus vesiculosus* (*F. vesiculosus*) showed high inhibitory activity against galectin-4 [14]. The glycan structure and molecular weight of natural fucoidans vary depending on the type of algae from which they are derived, and the extraction method also results in uneven molecular weight and structure [15]. The heterogeneity and non-uniformity of the structure and sulfation patterns of natural fucoidans make it difficult to study the precise structure–activity relationship for its binding to galectin-4. On the other hand, the synthesized fucoidan analogs have uniform, well-defined structures and sulfation patterns.

This study aims to clarify the structure–activity relationship in the suppression of peritoneal metastasis through systematic evaluation of synthetic fucoidan analogs with well-defined structures.

## 2. Results

### 2.1. Evaluation of the Inhibitory Activity of Natural Fucoidan Toward Galectin-4 and Cell Proliferation

The inhibitory activity towards galectin-4 binding was evaluated using a competitive enzyme-linked immunosorbent assay (ELISA) method. First, the inhibitory activity of natural fucoidan from *F. vesiculosus* against galectin-4 was measured using a cholesterol 3-sulfate-coated ELISA plate. Fucoidan inhibited the binding of galectin-4 in a dose-dependent manner and the half-maximal inhibitory concentration of the fucoidan was approximately 3 μg/mL (Figure 1A). Although fucoidan has a heterogeneous molecular weight, assuming it to be 50 kDa, 3 μg/mL is equivalent to 0.06 μM. The binding affinity of fucoidan to galectin-4 was also measured using surface plasmon resonance (SPR) by immobilizing galectin-4 on a sensor chip (Figure 1B). The dissociation constant (K_D_) value of fucoidan was tentatively determined as 2.25 × 10^−8^ by single-cycle kinetics. This value is considerably lower than those measured for carbohydrates, including lactose and GalNAcα1→3(Fucα1→2)Galβ1→4Glc, in a previous study [7]. Next, we studied the effect of fucoidan on the proliferation of MKN45 cells, a widely used human gastric cancer cell line that expresses galectin-4 and is derived from a poorly differentiated adenocarcinoma [16]. Cell proliferation was assessed using an adenosine triphosphate (ATP)-based cell viability assay. Natural fucoidan inhibited the proliferation of MKN45 cells in a concentration-dependent manner (Figure 1C). We observed that suppression of galectin-4 expression inhibited the expression of c-MET, hepatocyte growth factor receptor that has tyrosine kinase activity and phospho-c-MET (pMET), caused by autophosphorylation of c-MET [2]. Therefore, we expected that fucoidan would have the same effect as suppression of galectin-4 expression. Natural fucoidan suppressed c-MET and pMET (Figure 1D), similar to the effects of galectin-4 suppression [2].

Natural fucoidan also inhibited the proliferation of NUGC4 cells, a poorly differentiated gastric cancer cell line highly expressing galectin-4 [2], in a concentration-dependent manner (Figure 1E), as did MKN45 cells. We also examined whether knockout (KO) of galectin-4 affects the inhibitory effect of natural fucoidan. The inhibitory effect of fucoidan was weaker in KO cells than in wild-type (WT) cells, suggesting the involvement of galectin-4 in fucoidan’s inhibition mechanism (Figure 1E). The galectin-4 binding and growth inhibitory activities of natural fucoidan were both attenuated in the presence of fetal calf serum (FCS) (Appendix A), suggesting that a component of the serum inhibited interaction. These results imply that a galectin-4-mediated pathway may exist through which natural fucoidan inhibits the proliferation of gastric cancer cells.

However, due to the high molecular weight and heterogeneous structure of natural fucoidan [15], it is difficult to determine the precise structure that strongly binds to galectin-4. Hence, further experiments were performed using the synthesized fucoidan analogs with distinct structures.

### 2.2. Chemical Structures of the Fucoidan Analogs and Their Inhibitory Activity

#### 2.2.1. Chemical Structures of the Fucoidan Analogs

Fucoidans can be classified into three groups based on differences in their sugar backbone chains [8,17]. The structures of the fucoidan analogs used are shown in Figure 2. Type I, Type II, and Type III fucoidan groups are composed of repeating α(1,3)-linked, α(1,3)- and α(1,4)-linked, and α(1,4)-linked l-fucose, respectively (Figure 2A). Our strategy was to first select the most potent fucoidan analog from fucoidan analogs **1**–**13** [17]. The fucoidan analog **14** (Figure 2B) [18], comprising the cholestanyl group as the aglycone moiety instead of the octyl group in **10** was also studied, because cholestanol (Cho)-conjugation could potentially increase inhibitory activity.

#### 2.2.2. Inhibitory Activity of the Fucoidan Analogs

We first measured the inhibitory activities of all synthesized fucoidan analogs **1**–**14** to narrow down the candidate. Fucoidan analogs **2**, **3** and **14** showed rather stronger inhibitory activities on galectin-4 binding toward cholesterol 3-sulfate (Appendix A) and asialofetuin (Appendix A). Next, the inhibitory activity of fucoidan analogs on the proliferation of MKN45 cells was measured (Figure 3A). Fucoidan analogs **3** and **10** suppressed cell proliferation, and **14** showed the strongest growth inhibitory activity. In contrast, no significant inhibition was observed with the addition of Cho (100 μM) (Figure 3A). The inhibitory activities of fucoidan analogs **11** and **12** were comparable, we selected **12** as the control, because it has the same sulfation pattern (2-*O*-sulfated) with the fucoidan analog **3**, and natural fucoidan extracted from *F. vesiculosus* is mainly 2-*O*-sulfated [19]. From the viewpoint of growth and binding inhibitory activities, we narrowed it down to **2**, **3**, **10**, **12**, and **14** for further investigation.

Fucoidan analogs inhibited the proliferative ability of NUGC4 cells, but minimal inhibitory activity was observed against KO cells (Figure 3B). **14** showed strong growth inhibitory activity against NUGC4 cells at a concentration of 50 μM and the inhibitory activity was attenuated against KO cells in the lower concentration range (Figure 3C). These findings suggest that galectin-4 is at least partially responsible for growth inhibition by fucoidan analogs.

We also measured the inhibitory activity of fucoidan analogs on the proliferation of HEK293 cells (Appendix A). Little inhibitory activity was observed at the tested concentrations, suggesting that the inhibitory activity against gastric cancer cells was not due to simple cytotoxicity.

### 2.3. Properties of Fucoidan Analogs

#### 2.3.1. Fucoidan Analogs Inhibit the Galectin-4 Binding

Blood group A type 1 glycans exhibit a strong affinity for both galectin-4 [20,21] and SARS-CoV-2 [22]. We measured the inhibitory activity of the fucoidan analogs against galectin-4 using blood group A type 1 glycan as a ligand and found that **3**, **10**, **12**, and **14** effectively inhibited the galectin-4 binding to the ligand in a dose-dependent manner (Figure 4A). Among the tested fucoidan analogs, **14** showed the strongest inhibitory activity, which correlated with the results showing its strongest growth inhibitory activity against the cancer cells.

#### 2.3.2. Downregulation of pMET by Fucoidan Analogs

The expression levels of c-MET and pMET were studied to elucidate the growth inhibitory mechanism by fucoidan analogs, because downregulation of pMET was found in cells treated with natural fucoidan (Figure 1D) and by the knockdown of galectin-4 [2]. Distinct suppression of c-MET and pMET was observed in **3**-, **10**-, and **14**-treated cells in both MKN45 and NUGC4 cells (Figure 4B). Real-time polymerase chain reaction was performed to examine whether the reduction in c-MET in fucoidan analog-treated cells occurred at a genetic level. The relative genetic expression level of c-MET did not change in treated cells, suggesting that the reduction occurred by post-translational effect. These results are consistent with the finding that the reduction in c-MET and pMET occurred by the suppression of galectin-4 [2].

#### 2.3.3. Caspase-3 Activation Was Not Observed in Cells Treated with Fucoidan Analogs

Fucoidan analog **7** possessed apoptosis-inducing activity in breast cancer MCF-7 cells [23]; therefore, we investigated the apoptosis-inducing activity of fucoidan analogs against gastric cancer cells. We examined the cleaved-caspase-3 (CC-3) to detect apoptosis using PHA-665752 as a positive control [24]. PHA-665752, a selective small-molecule inhibitor of c-MET [25], increased CC-3 staining in MKN45 (Figure 4C-2) and NUGC4 cells (Appendix A). However, CC-3 staining was not observed in either MKN45 or NUGC4 cells following **10** and **14** treatments, suggesting that activation of caspase-3 did not occur in treated cells at the tested concentrations (Figure 4C and Appendix A).

### 2.4. SPR Reveals the Binding Behaviors of Fucoidan Analogs Toward Galectin-4

To directly evaluate the affinity of the fucoidan analogs for galectin-4, we used a galectin-4-immobilized SPR assay system. The synthesized fucoidan analogs **1**–**14** are linear fucose tetrasaccharide moieties with different glycosidic linkages and sulfation patterns. Lacto-*N*-tetraose (LNT) is also a tetrasaccharide composed of four monosaccharide units joined by β(1-3)-linkages in a linear chain, and we measured the K_D_ of LNT toward galectin-4 in our previous study [7]. Accordingly, we used LNT as a control in the SPR study. We first injected LNT to confirm whether galectin-4 retained its carbohydrate-binding activity after immobilization. LNT showed a rapid increase in response, followed by a rapid decrease to baseline after the injection was completed (Figure 5A), and the equilibrium binding constant was calculated as 2 × 10^−4^ M. We analyzed their bindings to galectin-4 using the single-cycle kinetics method by adding serial-diluted fucoidan analogs from 3 to 50 μM, because it was expected to be difficult to regenerate the surface after the injections of fucoidan analogs. As expected, some of the fucoidan analogs slowly dissociated from galectin-4 on the sensor chip after the injection was completed (Figure 5). An accurate K_D_ could not be determined because it was difficult to apply high concentrations of fucoidan analogs owing to solubility concerns. Instead, we compared each sensorgram to evaluate the relative binding ability to galectin-4. Fucoidan analogs **2**, **3**, **10**, **12**, and **14** bound to galectin-4 dose-dependently (Figure 5B–E,G). Fucoidan analogs **4**, **9**, **13**, and Cho did not show distinct binding (Appendix A and Figure 5F). Notably, **14** showed a significantly increased response during the association phase (Figure 5G). The unstable sensorgram observed during the association phase of fucoidan analog **14** may be due to non-specific binding of its aglycone component, Cho. The highest RU in the association phase of **14** was 1085, exceeding that of **10** by more than 70-fold. The molecular weight of **14** (1892) was slightly larger than that of **10** (1633). Therefore, this difference in RU values was not due to molecular weight, suggesting that **14** may become multimerized when bound to galectin-4. To confirm this, we evaluated the potential for micelle formation of **14**, which is an amphiphilic molecule bearing a hydrophobic sterol moiety at the reducing end and may be aggregate to form micelles in an aqueous environment. The existence of **14** as a monomer or in a micelle form under physiological conditions is indicated by its critical micelle concentration (CMC), a key parameter that can influence its pharmacological activity. The CMC value of **14** is measured by fluorescent spectrophotometry using the probe, *N*-phenyl-1-napthylamine (NPN) [26]. As depicted in Appendix A, the CMC value of **14** was about 100 μM, which was higher than the concentrations in the SPR. These results may suggest that the aggregation of **14** was promoted on the galectin-4-immobilized sensor chip.

### 2.5. Molecular Docking of 10 and 14 with Galectin-4N Domain

The screening strategy was further prolonged with molecular docking simulation to ascertain the potential binding sites of fucoidan analogs **10** and **14** on the surface of the human galectin-4N terminal carbohydrate recognition domain (N-CRD), protein data bank identifier (PDB ID): 5DUW. In the crystal structure of 5DUW–glycerol complex, the galectin-4N binding site adopts a pre-organized conformation favorable for saccharide recognition [27]. Initially, we identified all the distinct binding poses of 5DUW–**10** and 5DUW–**14** complexes, each representing a unique orientation and position of both **10** and **14** within the protein pocket. At their optimal binding poses, both analogs demonstrated effective binding to 5DUW, with maximum binding energies of -6.1 kcal/mol (Appendix A) for **10** and −7.1 kcal/mol (Appendix A) for **14**. Notably, analog **14** exhibited superior binding to 5DUW compared to analog **10**.

The interactions between 5DUW binding sites with **10** and **14** were illustrated in the 2D interaction diagrams (Figure 6A,B). Previous studies have revealed that 5DUW recognizes the sulfate cap of 3’-sulfated glycans through a crucial interaction with Arginine-45 (Arg45), a residue that plays a central role in the protein’s binding mechanism due to its ability to form electrostatic interactions, strategic positioning within the protein, and contribution to enhanced affinity and specificity [13,27]. The interactions within the 5DUW–**10** complex highlight various types of non-covalent forces that contribute to binding affinity and stability. Conventional hydrogen bonds (green lines) between the sulfate groups of **10** and residues such as Lys44, Arg45, Arg67, Trp71, Lys73, Trp84, Arg89, and Arg91 play a crucial role in stabilizing the complex. In addition, Arg45 is involved in conventional hydrogen bonding with ring oxygen of fucose further augmenting the complex stability (Figure 6A). π-related interactions, including π-anion (orange lines) and π-sulfur (yellow lines), are observed with residues such as Arg45, Trp84, and Arg89 indicating significant aromatic and electrostatic contributions. Alkyl and π-alkyl interactions (pink lines) with residues like Phe47, Val51, and Val60 further reinforce the hydrophobic environment. Additionally, salt bridges (orange lines) with Arg45, Arg67, and Lys73, and attractive charge interactions (orange lines) involving Arg89, and Arg91 enhance the anchoring of the ligand within the binding site.

**14** exhibited a stronger binding profile with 5DUW as evidenced by diverse non-covalent interactions. The 2D interaction diagram reveals multiple conventional hydrogen bonds (green lines) with key residues such as Arg45, Asn49, Gly57, His63, Asn65, Trp84, and Gln137 which contribute to the complex, 5DUW–**14** stabilization (Figure 6B). The presence of attractive charge interaction (orange line) with Arg45 provides significant electrostatic contributions to the binding. Additionally, π-related interaction like π-anion (orange line) with Phe47 further enhances the aromatic contributions. π-sulfur interactions (yellow) with residues such as Tyr20, Phe47, His63, and Trp84 reinforce the non-polar contributions to binding thereby stabilizing the complex through enhanced hydrophobic interactions. Arg45 plays a crucial role in stabilizing both complexes through their involvement in both hydrogen bonding and electrostatic interactions as aligned with the previous reports [27]. Specifically, it forms conventional hydrogen bonds (green lines) with both **10** and **14**, enhancing the overall stability of the interaction and participating in attractive charge interactions (orange lines).

By comparing the stability of both 5DUW–**10** and 5DUW–**14** complexes, it is evident that the stabilization of both is driven by extensive non-covalent interactions. Despite sharing the same glycan skeleton, the two ligands exhibit distinct binding profiles, primarily due to differences in their aglycone moieties. The *n*-octyl group in **10** is a linear, flexible, and small hydrophobic tail, whereas the cholestanyl group in **14** is bulky, rigid, and highly hydrophobic. However, the 2D interaction diagrams reveal that hydrophobic interactions between the cholestanyl group and 5DUW are more limited in the 5DUW–**14** complex compared to the 5DUW–**10** complex (Figure 6A,B). This is due to the steric factor of the cholestanyl group indirectly enhances binding by positioning the glycan in a way that optimizes electrostatic interactions with sulfated residues of **14** and 5DUW. This suggests that the cholestanyl group acts as a conformational anchor, ensuring that the glycan remains in an orientation that maximizes favorable electrostatic interactions with the protein. Consequently, **14** emerges as a more effective modulator of 5DUW, demonstrating superior binding stability and interaction efficiency.

To validate the reliability of our docking protocol, we conducted a redocking study using the co-crystallized ligand, lactose-3′-sulfate, applying the same docking parameters used for our test compounds. The ligand was redocked into the binding site, and the resulting pose was compared with the crystallographic conformation. The redocking closely reproduced the experimental pose with a root-mean-square deviation (RMSD) of 0.62 Å (Appendix A) and a docking energy of −7.0 kcal/mol. The low RMSD value demonstrates a close agreement between the docked and crystallographic poses, confirming that our docking procedure, including the chosen parameters, grid definition, and preparation protocols, accurately captures the key interactions and binding orientation of the ligand.

### 2.6. Arg45 of Galectin-4 Is Important for Binding to Fucoidan Analogs

Arg45 was suggested to be important for interaction with fucoidan analogs by molecular docking, and our previous study showed that Arg45 of galectin-4 is important for binding to sulfated compounds [14]. Therefore, we generated a mutant galectin-4 with Arg45 replacing Ala45 (R45A) and evaluated its binding ability to fucoidan analogs. The binding response of **3** and **12** to the R45A mutant was significantly reduced compared to that of the WT galectin-4 (Figure 7A and Appendix A). In contrast, the binding response of **10** and **14** to the R45A mutant was not significantly reduced compared to that of WT galectin-4. These results suggested that **10** and **14** interact with galectin-4 differently from **3** and **12**.

Next, we assessed their ability to bind to galectin-4N- and C-CRDs. The importance of the binding of **3** and **12** to Arg45 in N-CRD can be inferred from the fact that little binding to C-CRD was observed (Figure 7A and Appendix A). **10** and **14** bound to both N- and C-CRDs (Figure 7A and Appendix A). These results were likely attributed to the different sulfation patterns of the fucoidan analogs, **10** is 2,3-*O*-sulfated with 9 sulfate groups, **3** and **12** are 2-*O*-sulfated with 5 and 4 sulfate groups, respectively (Figure 7B).

The SPR study inferred the multimerization of **14** (Figure 5G). Next, a sedimentation assay was conducted to further investigate the interaction between galectin-4 and **14** since we already reported that galectin-4 can be precipitated by lipids with affinity [14]. The WT galectin-4 precipitated with **14**, showing a band at the position corresponding to the dimer; however, very few R45A mutants were precipitated, suggesting that Arg45 is important for strong binding to **14** (Figure 7C).

## 3. Discussion

While numerous studies have highlighted the anticancer effects of fucoidan, including its role in inducing apoptosis, inhibiting tumor growth, suppressing metastasis, and modulating key signaling pathways, only a few have provided direct evidence of its molecular interactions with specific ligands [9,28,29,30]. To the best of our knowledge, this is the first study to demonstrate the involvement of galectin-4 in the suppressive effects of fucoidan on malignant gastric cancer cells.

Mak et al. have shown that the growth inhibitory activity of natural fucoidans varies among different cell types, indicating that their effects are cell-specific rather than not just imparting cytotoxicity [31]. Galectin-4-expressing colon cancer cells were inhibited relatively at low fucoidan concentrations, supporting that galectin-4 is involved in the suppressive effects of fucoidan [31]. The difference in growth inhibition between WT and galectin-4 KO cells partially supports the above idea; however, fucoidan may exert its effects through additional pathways beyond galectin-4-mediated signaling.

Non-galectin-4 specific cytotoxic effects can be investigated by using cells that lack galectin-4 expression. Growth suppression was observed in NUGC4 KO cells, which lack galectin-4 expression, with natural fucoidan and fucoidan analog **14**, suggesting non-galectin-4-specific cytotoxic effects. Natural fucoidans induce apoptosis in various cancer cells through both the intrinsic mitochondrial pathway and the extrinsic death receptor pathway, often involving the activation of caspases, altered Bcl-2/Bax ratios, and the modulation of MAPK/ERK and PI3K/AKT signaling pathways [12]. Fucoidan analog **14** did not activate caspase-3 at the low concentrations tested. However, at high concentrations, the apoptosis pathway reported for natural fucoidan may be activated, necessitating further investigation.

The lack of inhibitory activity of fucoidan analogs **4**, **9**, and **13** was consistent with the results of our previous study, which showed that sulfate groups are important for galectin-4 binding [7,13]. In addition, **1**, **5**, and **10** have 9 sulfate groups in each molecule, but their binding abilities toward galectin-4 are different, suggesting that not only the degree of sulfation but also the glycan linkage are important for binding. This was also supported by the fact that none of the fucoidan analogs in the Type II group showed high inhibitory activity. Furthermore, based on molecular docking simulation and mutagenesis studies using SPR, Arg45 in galectin-4 is important and contributes to the binding of galectin-4 to fucoidan analogs. 

The inhibitory activities of fucoidan analogs **1**–**13** against heparin binding to the SARS-CoV-2 spike (S) protein have been evaluated in our previous study [17]. Notably, galectin-4 and SARS-CoV-2 receptor binding domain have similar β-sheet structures, and both show high affinities toward blood group A type 1 antigen [32,33]. Fucoidan analog **10**, which had the strongest inhibitory activity against the SARS-CoV-2-S among our synthesized fucoidan analogs [17], also showed strong inhibitory activity against galectin-4. Furthermore, our recent study revealed that **14**, a Cho-conjugated form of **10**, significantly increased the affinity for the SARS-CoV-2-S [18]. Our strategy was to first select the most potent one from fucoidan analogs **1**–**13**. We had planned to conjugate with Cho if any of fucoidan analogs **1**~**13** proved more promising than **10**. However, none of the fucoidan analogs (including **3**) surpassed **10**, particularly concerning proliferation-inhibiting activity (Appendix A).

The predicted impact of Cho-conjugation can be evaluated by comparing the activities of fucoidan analogs **10** and **14**. Chemically synthesized sugar, Cho inhibits the proliferation of colorectal and gastric cancer cells [34]. Although there are differences between their study and ours in the method of administration to cells, they showed that the Cho-conjugated compounds were rapidly taken up via lipid rafts/microdomains on the cell surface and altered the expression levels of apoptosis-related molecules in cancer cells. High-molecular-weight natural fucoidan exhibits high inhibitory activity even when part of its structure binds to galectin-4. It is interesting to mention that **10** and **14** possess the same glycone moiety that binds to galectin-4. However, **14** exhibited higher growth inhibitory and galectin-4 binding inhibitory activities than **10** did. The large SPR response suggests multimerization of **14**, which may also lead to a significant increase in the inhibitory activity due to steric inhibition (Figure 8A). The weak binding interaction between lectins and carbohydrate moieties can be compensated by the multivalent display of glycans [35,36]. In particular, multivalent ligands substantially enhance binding to galectin-4 [37]. Collectively, conjugation with Cho may synergistically enhance the anticancer activity of fucoidan analogs.

Fucoidans may also be ligands for other galectins. As shown in Appendix A, NUGC4 cells mainly express galectin-3 and -4. Unlike galectin-4, there is no correlation between galectin-3 expression and the peritoneal dissemination ability of gastric cancer cells [2]. Therefore, even if galectin-3 is inhibited by fucoidan, it is thought that this does not relate to the suppression of peritoneal dissemination.

Characterization of the absorption, distribution, metabolism, and excretion (ADME) properties is important for drug administration. However, such experimental studies are lacking for galectin-4. Generally, surface charge mediates interactions with various cellular and extracellular components, while mass affects diffusion and clearance mechanisms [38]. The molecular weight of galectin-4 is approximately 36 kDa, such a smaller protein has faster tissue penetration but more rapid elimination, leading to a shorter duration in the circulation. Furthermore, galectin-4 may bind to specific glycoproteins or glycolipids in serum; this is inferred from the fact that both galectin-4 binding and growth inhibitory activities of natural fucoidan were attenuated in the presence of serum. Structural study revealed that positively charged residues are located on the surface of galectin-4 [39]. Galectin-4 is expressed mainly in the epithelial cells of the gastrointestinal tract, and its surface charge facilitates its interactions with cells and tissues, forming complexes with other membrane proteins and lipids. Metabolism of galectin-4 is influenced by interactions with cellular components, such as lipid rafts, and is part of its functional role. Considering all of this background, intraperitoneal administration may be preferable to intravenous administration.

Galectins are known to be involved in the regulation of various molecules by the formation of so-called galectin-lattices through cross-linking their ligand molecules at the cell surface [40]. For example, galectin-9 induced proliferation of human osteoblasts by clustering of lipid rafts on the cell surface [41], and galectin-3 cross-linked *N*-glycans on epidermal growth factor and transforming growth factor receptors at the cell surface, delaying their removal by constitutive endocytosis [42]. Galectin-4 also has been reported to facilitate stabilization of lipid rafts through the formation of lattices with some glycoproteins and glycolipids [3,43]. As shown in Figure 8B, galectin-4 may also be involved in signal regulation by stabilizing c-MET and other molecules on the cell surface (manuscript in preparation). However, in the absence of galectin-4 or loss of binding ability, the function of galectin-4 in signaling is considered to be weakened.

In conclusion, our results demonstrated that fucoidan analog **10**, with a Type III structure containing repeating 2,3-*O*-sulfated α(1,4)-linked L-fucose, exhibited significant suppressive activity against the proliferation of malignant gastric cancer cells via galectin-4 inhibition. Fucoidan analog **14**, in which the octyl group of **10** was changed to a cholestanyl group, showed the strongest suppressive activity, confirming that the aglycon structure was important for binding to galectin-4. However, further detailed analyses are required to fully understand the precise mechanisms involved in fucoidan analog-mediated regulation of gastric cancer cells. More effective modifications may be required to improve its therapeutic effect in vivo. Overall, our findings indicate that fucoidan analog **14** is a promising lead compound for treating gastric cancer metastasis. 

## 4. Materials and Methods

### 4.1. Materials and Antibodies

Cholesterol 3-sulfate, asialofetuin, fucoidan from *F. vesiculosus* (F5631, Lot #SLCD5199), bovine serum albumin (BSA) (A-2058), and polyvinylidene difluoride (PVDF) membranes were purchased from Merck (Darmstadt, Germany). NPN and 4% paraformaldehyde (PFA) phosphate-buffer solution, and L-alanyl-L-glutamine solution were purchased from FUJIFILM Wako Pure Chemical (Osaka, Japan). β-cholestanol was purchased from Tokyo Chemical Industry Co. (Tokyo, Japan). Skimmed milk and 1,10-phenanthroline were purchased from Nacalai Tesque, Inc. (Kyoto, Japan).

Horseradish peroxidase (HRP)-conjugated anti-rabbit IgG (#65-6120) and anti-mouse IgG (H + L) (#62-6520) antibodies were obtained from Thermo Fisher Scientific (Waltham, MA, USA). Anti-c-MET (#8198), and anti-phospho-c-MET (Tyr1234/1235, #3077) antibodies were purchased from Cell Signaling Technology (Danvers, MA, USA). Goat anti-galectin-4 antibody (AF1227) was purchased from R&D Systems (Minneapolis, MN, USA). Anti-β-actin mAb (66009-1-Ig) was purchased from Proteintech (Rosemont, IL, USA).

### 4.2. Galectin-4 Preparation

Human recombinant galectin-4, R45A mutant, and galectin-4-N- and -C-domains were prepared as previously described with some modifications [14]. Briefly, the pQE-9 plasmid (Qiagen, K.K., Tokyo, Japan) containing full-length galectin-4, R45A mutant, and galectin-4-N-domain was transformed into the *E. coli* strain M15[pREP4]. *E. coli* transformants were cultured, and proteins were induced by the addition of isopropyl-β-D-thiogalactopyranoside (IPTG) to the final concentration of 0.5 mM according to the manufacturer’s instructions. From the lysate of the harvested culture, His-tag proteins were purified using TALON Metal Affinity Resin (Takara Bio, Inc., Yamanashi, Japan) according to the manufacturer’s instructions. Recombinant galectin-4 C-domain was prepared from the glutathione-S-transferase (GST) fusion galectin-4-C-domain, which was prepared as described previously [44]. In brief, pGEX-6P-1 plasmid (Cytiva, Tokyo, Japan) containing galectin-4 C-domain was transformed into the *E. coli* strain BL21. *E. coli* transformants were cultured, and proteins were induced by the addition of IPTG to a final concentration of 0.5 mM. From the lysate of the harvested culture, the GST-tagged C-domain was purified using glutathione-Sepharose (Cytiva). GST was removed from the GST-C-domain by PreScission Protease (Cytiva) according to the manufacturer’s instructions. Their lectin activity was evaluated by measuring their ability to bind asialofetuin. 

### 4.3. Cell Lines and Cell Culture

Human gastric cancer cell lines, MKN45 and NUGC4, were obtained from the RIKEN BioResource Center (Tsukuba, Japan). Galectin-4 KO cells were established as previously described [2]. Galectin-4 KO cells were established by CRISPR/Cas9-mediated genome editing using the Dharmacon Edit-R CRISPR-Cas9 platform (Horizon Discovery Ltd., Cambridge, UK) as previously described [2]. Briefly, cells were transfected with trans-activating CRISPR RNA, human galectin-4 CRISPR RNA, and Cas9 expression plasmid. Stable KO clones were selected after puromycin (0.5 μg/mL) selection. The absence of galectin-4 expression was verified by Western blot. Cells were cultured in RPMI 1640 (Thermo Fisher Scientific, Tokyo, Japan) medium supplemented with 2 mM L-alanyl-L-glutamine solution and 10% FCS.

### 4.4. Cell Proliferation Assay for Growth Inhibition Experiments

For growth inhibition experiments, MKN45 or NUGC4 cells were grown in 96-well plates at 5 × 10^3^/well on the previous day, and the cell supernatant was discarded and washed once with phosphate-buffered saline (PBS). Fucoidan or fucoidan analog solutions diluted in FCS-free Advanced RPMI 1640 medium were added to the cells and cultured for another 3 days. Cell proliferation was measured using the ATP-based cell viability assay.

An ATP-based cell viability assay was performed to determine the number of viable cells in culture by quantifying the amount of ATP present. CellTiter-Glo 2.0 reagent (Promega, Madison, WI, USA) equal to the volume of the culture medium in each well was added to the cells and mixed on a shaker for 2 min. The plates were incubated at room temperature for 10 min and luminescence was measured using a luminometer (TriStar2 LB942; Berthold Technologies, Wildbad, Germany). The average values of triplicate wells are shown.

### 4.5. ELISA for Binding of Galectin-4

ELISAs for the binding of galectin-4 were performed as described previously with some modifications [14]. Briefly, cholesterol 3-sulfate (1 μg/well) or blood group A type 1 tetrasaccharide-neoglycolipid (L241, Dextra Laboratories, Reading, UK) (20 pmol/well) in 10 μL of MeOH were added to each well of a 96-well microtiter plate (Nunc-Immuno Plate Maxisorp Surface, Merck). After evaporation of the solvent, 100 μL of 1% BSA in PBS was added as a blocking solution, and the plate was left overnight at 4 °C. After washing with PBS, 50 μL of galectin-4 in the blocking solution were added to each well, and the plate was left for 1 h at room temperature. The plate was washed several times with washing buffer (0.01% Tween 20 in PBS), and anti-His-tag mAb-HRP-DirecT (D291-3; MBL, Tokyo, Japan) diluted in the washing buffer was added and incubated for 1 h at room temperature. After washing, the plate was washed and incubated with tetramethylbenzidine solution (Thermo Fisher Scientific), and the released chromogen was measured at 450 nm using a photo spectrometer (SH-1300Lab, Corona Electric, Hitachinaka, Japan).

### 4.6. ELISA-Based Inhibition Assay

An inhibition ability assay against galectin-4 binding was evaluated by ELISA-based inhibition assay. Fifty μL of galectin-4 (0.25 μg/mL) with or without various concentrations of fucoidan analogs in 1% BSA in PBS were applied to ligand-coated plates, and ELISA was performed as described above. The inhibitory ability was evaluated by comparing the data of wells with inhibitors to the data of wells without inhibitors.

### 4.7. SPR Measurements

Direct interaction between galectin-4 and inhibitors (natural fucoidan or fucoidan analogs) was analyzed by SPR. The binding of inhibitors results in changes in refractive index on the galectin-4 immobilized sensor chip. The binding of the analyte (inhibitors) is measured; there is an increase in mass associated with the binding event. An increase in mass leads to a proportional increase in the refractive index, resulting in a change in response. SPR binding experiments were performed on a Biacore X100 instrument (Cytiva) at 25 °C in HBS-EP buffer (BR100188, Cytiva, 0.01 M 4-(2-hydroxyethyl)-1-piperazineethanesulfonic acid pH 7.4, 0.15 M NaCl, 3 mM ethylenediaminetetraacetic acid (EDTA), 0.005% *v*/*v* Surfactant P20). Galectin-4 was immobilized on a research-grade CM5 sensor chip at pH 5.5, and the amount of galectin-4 immobilized by amine coupling was 9000–11,000 RU. Analytes in HBS-EP buffer were introduced onto the surface at a flow rate of 30 µL/min. Binding studies consisted of the injection (association 180 s, dissociation 250 s) of various concentrations of inhibitors (1.25–20 µg/mL for fucoidan, 3.15–50 μM for fucoidan analogs) in the single cycle mode. LNT (31.3–500 μM) was injected in the multi cycle mode. The relative response was determined by subtracting the blank values obtained on the non-immobilized surface from the values obtained on the galectin-4-immobilized surfaces. The data were evaluated using Biacore X100 evaluation software (version 2.01). We calculated the tentative K_D_ value of natural fucoidan from *F. vesiculosus,* assuming a tentative molecular weight of 50 kDa.

### 4.8. Western Blot Analysis

Cells were washed once with PBS and scraped into SDS sample buffer (62.5 mM Tris-HCl pH 6.8, 2% SDS [*w*/*v*], and 25% glycerol [*v*/*v*]) or lysis buffer (20 mM Tris-HCl, 150 mM NaCl, 1 mM Na_2_EDTA, 1% Triton X-100, 1 mM Na_3_VO_4_, 1 mM phenylmethanesulfonyl fluoride, containing cOmplete Protease Inhibitor Cocktail and PhosSTOP, Merck). The samples were then heated to 100 °C for 5 min and sonicated for 20 s. Proteins were separated using 4–15% SDS-PAGE and transferred onto a PVDF membrane. After blocking with 5% skim milk in Tris-buffered saline with Tween 20 (TBS-T, 20 mM Tris-HCl, 150 mM NaCl, pH 7.5, containing 0.1% Tween 20), the membranes were incubated with primary antibodies at 4 °C overnight. After washing with TBS-T, the membranes were then incubated with an HRP-conjugated secondary antibody for 1 h at 25 °C. Immunoreactive proteins were visualized with a chemiluminescence-based procedure using Chemi-Lumi One Super Substrate (Nacalai Tesque, Inc.) and imaged using ChemiDoc XRS+ system (Bio-Rad, Hercules, CA, USA). The relative intensity of each band was determined using Image Lab software (Bio-Rad).

### 4.9. Synthesis of Fucoidan Analogs

Fucoidan analogs **1**–**14** were synthesized according to reported procedures [17,18].

### 4.10. Molecular Docking

The 3D structure of the target protein, human galectin-4N (PDB ID: 5DUW) was obtained from the RCSB PDB. The crystal structure of 5DUW is a tetramer comprising four identical chains, each co-crystallized with the ligand lactose-3′-sulfate, which also serves as the native ligand. For docking studies, a single chain containing the CRD was retained, while the remaining chains were removed. Protein preparation included the removal of non-essential water molecules, ions, and heteroatoms, the addition of polar hydrogens and charges, and modeling of missing residues to ensure a complete and stable structure. The chemical structures of compounds **10** and **14** were drawn in ChemDraw Ultra 14.0 and energy-minimized using OpenBabel embedded in PyRx-Python Prescription 0.8. Molecular docking was performed with AutoDock Vina embedded in PyRx-Python Prescription 0.8 using a 50 × 50 × 50 Å grid box centered to encompass all key active site residues [45]. For each ligand, 20 poses were generated, ranked according to binding affinity, and visually inspected. The top docking poses were selected based on lowest binding energies and preservation of critical interactions, and the resulting complexes were further analyzed with BIOVIA Discovery Studio Visualizer v24.1.0.23298 for 2D and PyMOL 3.0.3 for 3D visualization [46]. To validate the docking protocol, the co-crystallized ligand, lactose-3′-sulfate, was redocked into the binding site of 5DUW using the same parameters applied to our test compounds; however, a smaller grid box (25 × 25 × 25 Å) was used to accommodate the reduced size of the ligand and the active site. RMSD calculations were performed using BIOVIA Discovery Studio Visualizer v24.1.0.23298. Prior to calculation, the protein backbone was aligned to ensure proper superposition. RMSD values were computed based on heavy atoms of the ligand, excluding hydrogens. The overlay of the re-docked and crystallographic ligands is shown in Appendix A.

### 4.11. CMC Determination

NPN was recrystallized in hot methanol before use. The working solution of NPN was prepared at a concentration of 1 μM by diluting the NPN solution in methanol into HBS-EP buffer shortly before use. For the determination of CMCs, a series of vials with different concentrations of **14** were prepared in the buffer containing 1 μM NPN and maintained at the temperature of measurement for 1 h before the emission intensity was measured.

The uncorrected fluorescence emission spectrum was recorded on a JASCO FP-8500 spectrofluorometer (JASCO Corporation, Tokyo, Japan) using a 5 mm cuvette at 25 °C. For CMC determination, the excitation wavelength was set to 350 nm with a slit width of 5 nm. The emission intensity was recorded at the apparent maximum of the emission peak at 418 nm, also with a slit width of 5 nm.

### 4.12. Immunofluorescence Studies

Cells were fixed with 4% PFA solution for 20 min, permeabilized with 0.2% Triton X-100 in PBS for 5 min, and incubated with 1% BSA for 0.5 h at room temperature. For staining, cells were incubated with fluorescein isothiocyanate (FITC)-conjugated anti-cleaved-caspase-3 (No. 560901; BD Biosciences, Franklin Lakes, NJ, USA) in 1% BSA at room temperature for 1 h. After extensive washing, the cells were imaged using a confocal microscope (Fluoview FV10i; Evident, Tokyo, Japan).

### 4.13. Co-Sedimentation Assays

Co-sedimentation assays between galectin-4 and fucoidan analog **14** were per-formed to detect their interaction as previously described with some modifications [14]. Briefly, five nmol equivalents of **14** and Cho were dissolved in TBS (50 μL) and sonicated at 0 °C for 15 min to prepare lipid vesicles. TBS alone, **14** or Cho solutions (25 μL) were combined with 1.5 μg of galectin-4 (or R45A galectin-4 mutant). They were incubated at 37 °C for 30 min and then centrifuged at 15,000× *g* for 5 min. The supernatants were transferred to another tube, the pellets were subjected to SDS-PAGE, and the proteins were stained with Instant Blue Coomassie Protein Stain (ISB1L, Abcam, Cambridge, UK).

## Figures and Tables

**Figure 1 ijms-26-09228-f001:**
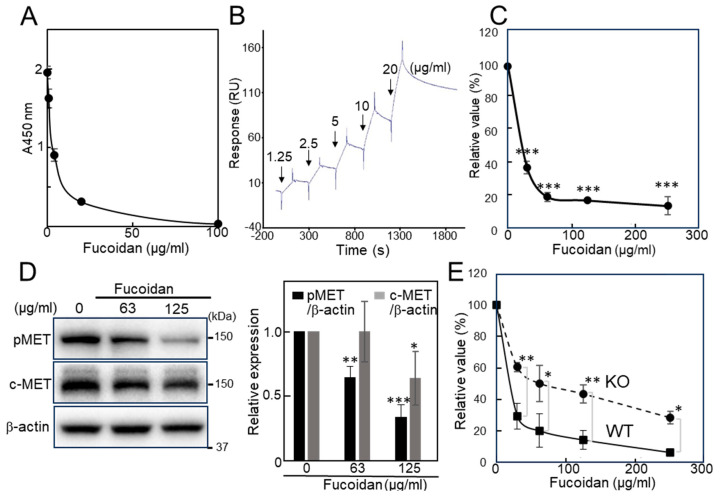
Effect of natural fucoidan. (**A**) ELISA-based inhibition assay of fucoidan for the binding of galectin-4 to cholesterol 3-sulfate. (**B**) SPR sensorgram of fucoidan toward galectin-4. The increasing concentrations of fucoidan were introduced to the galectin-4 immobilized surface and the relative response was determined by subtracting the blank values obtained on the non-immobilized surface from the values obtained on the galectin-4-immobilized surfaces. (**C**) ATP-based cell viability assay of fucoidan on MKN45 cells. (**D**) The Western blot analysis of fucoidan treated MKN45 cells with pMET, c-MET, and β-actin antibodies. The MKN45 cells were treated with fucoidan diluted in FCS-free Advanced RPMI 1640 medium for 3 days, and representative results are presented. Bar graphs represent pMET and c-MET expression normalized to that of β-actin from three independent experiments. (**E**) ATP-based cell viability assay of fucoidan on NUGC4 WT and KO cells. Cell proliferation was measured by ATP assay. The value from the ATP assay of the non-treated cells was set at 100%, and the relative value of treated cells was presented. Each value represents the mean ± SD from three independent experiments (**C**,**E**). Assuming the molecular weight of natural fucoidan is 50 kDa, 100 μg/mL is equivalent to 2 μM. Statistical analyses were performed using Student’s *t*-test. * *p* < 0.05, ** *p* < 0.01, *** *p* < 0.001; ELISA, enzyme-linked immunosorbent assay; SPR, surface plasmon resonance; RU, response unit; ATP, adenosine triphosphate; pMET, phospho-c-MET; FCS, fetal calf serum; WT, wild-type; KO, knockout.

**Figure 2 ijms-26-09228-f002:**
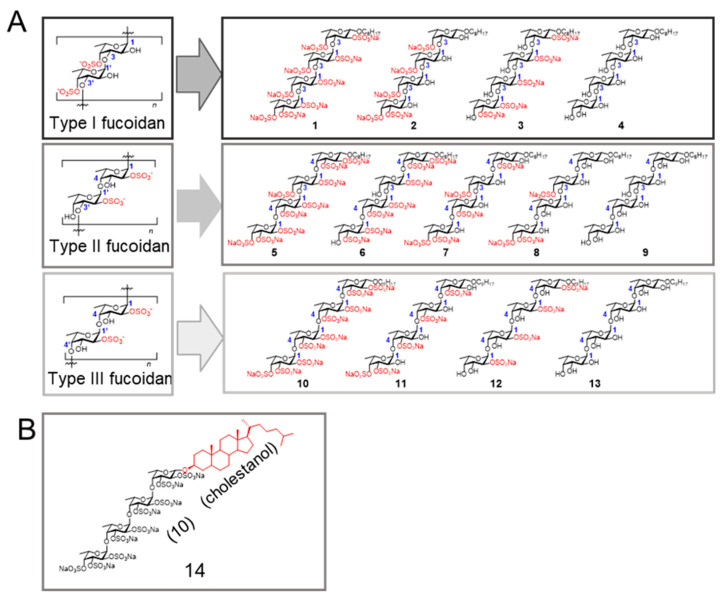
Chemical structures of fucoidan analogs (**A**) **1**–**13** and (**B**) **14** (cholestanyl group as the aglycone moiety instead of the octyl group in **10**).

**Figure 3 ijms-26-09228-f003:**
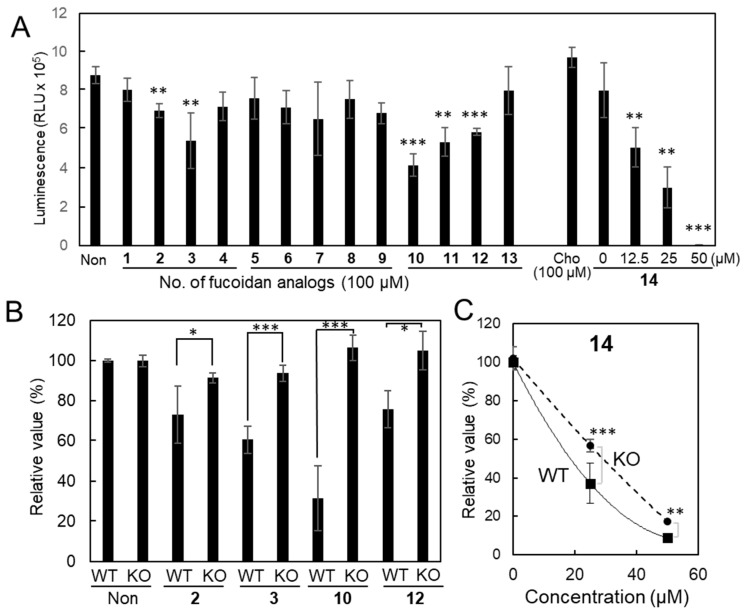
Effect of fucoidan analogs on the cell proliferation of gastric cancer cells. (**A**) MKN45 cells. **1**–**13** (100 μM) and **14** (12.5–50 μM). (**B**,**C**) NUGC4 WT and KO cells. **2**, **3**, **10**, **12** (50 μM) and **14** (25 and 50 μM). Cell proliferation was measured by an ATP-based cell viability assay. The value from the ATP assay of the non-treated WT cells was set at 100%, and the relative value of treated cells was presented (**B**,**C**). The mean ± SD of one representative experiment from three independent experiments is presented. Statistical analyses were performed using Student’s *t*-test. * *p* < 0.05, ** *p* < 0.01, *** *p* < 0.001. Non, non-treated; Cho, cholestanol; WT, wild-type; KO, knockout.

**Figure 4 ijms-26-09228-f004:**
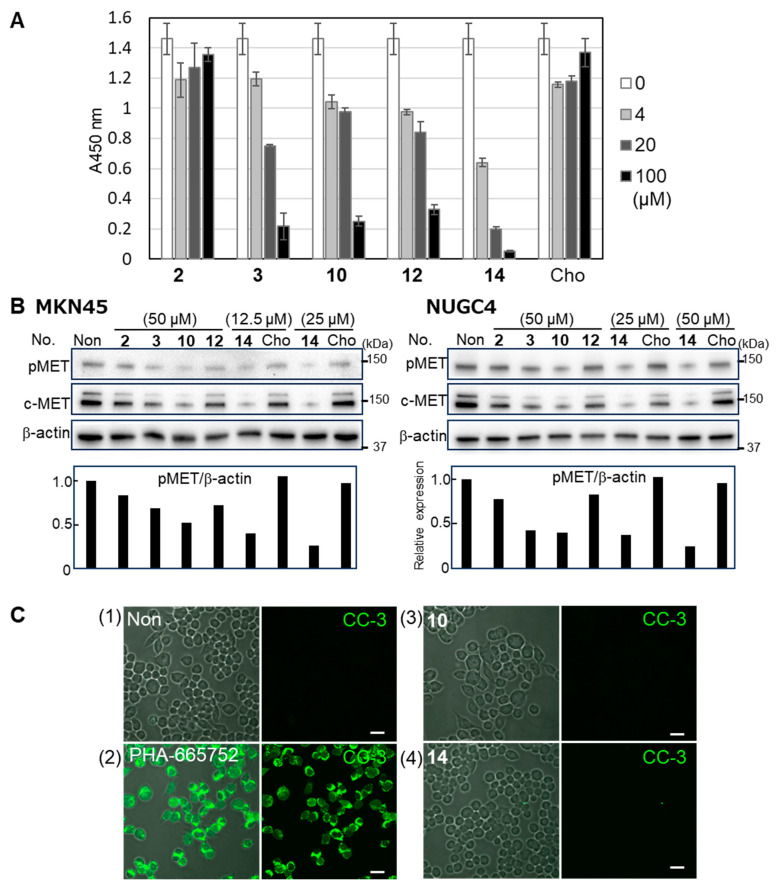
Effects of fucoidan analogs on galectin-4 binding, pMET and c-MET expression. (**A**) The inhibitory activity of fucoidan analogs against galectin-4 binding to blood group A type 1-neoglycolipid. Various concentrations of fucoidan analogs were incubated with galectin-4. The value from well without fucoidan analogs was set at 100%, and the relative values from the wells with fucoidan analogs were presented. (**B**) The Western blot analysis of fucoidan analogs-treated MKN45 and NUGC4 cells with pMET, c-MET, and β-actin antibodies. Bar graphs represent pMET expression normalized to that of β-actin. (**C**) Immunofluorescence labeling of the cleaved-caspase-3 (CC-3). MKN45 cells were incubated with (1) Advanced RPMI 1640 medium alone, (2) 2 μM PHA-665752, (3) 50 μM **10**, and (4) 25 μM **14** for three days. Cells were fixed and stained as described in Section 4. Stained images with anti-CC-3 and the merged confocal images with corresponding differential interference are shown (Scale bar, 20 μm). Non, non-treated; Cho, cholestanol; CC-3, cleaved caspase-3.

**Figure 5 ijms-26-09228-f005:**
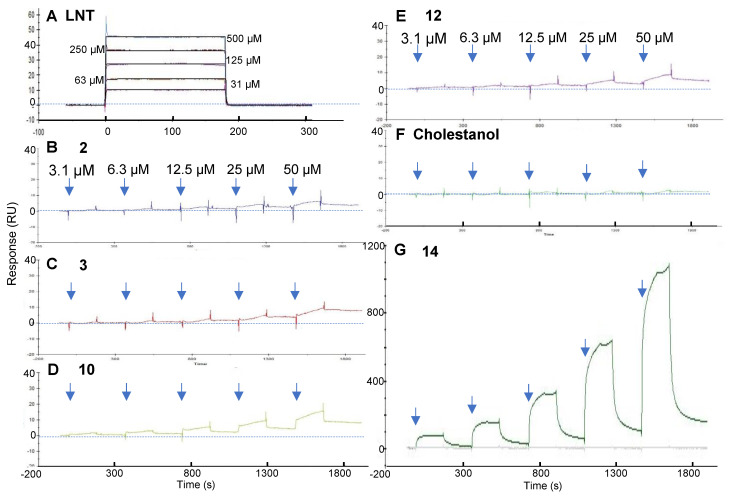
Sensorgrams of SPR for binding of fucoidan analogs to galectin-4. Increasing concentrations of (**A**) LNT, (**B**) **2**, (**C**) **3**, (**D**) **10**, (**E**) **12**, (**F**) Cholestanol and (**G**) **14** were introduced to the galectin-4 immobilized surface. The relative response was determined by subtracting the blank values obtained on the non-immobilized surface from the values obtained on the galectin-4-immobilized surface. The sensorgrams from the zero concentrations have been subtracted to correct for any systematic disturbances. The blue dotted line indicates zero RU. LNT, lacto-*N*-tetraose; RU, response unit.

**Figure 6 ijms-26-09228-f006:**
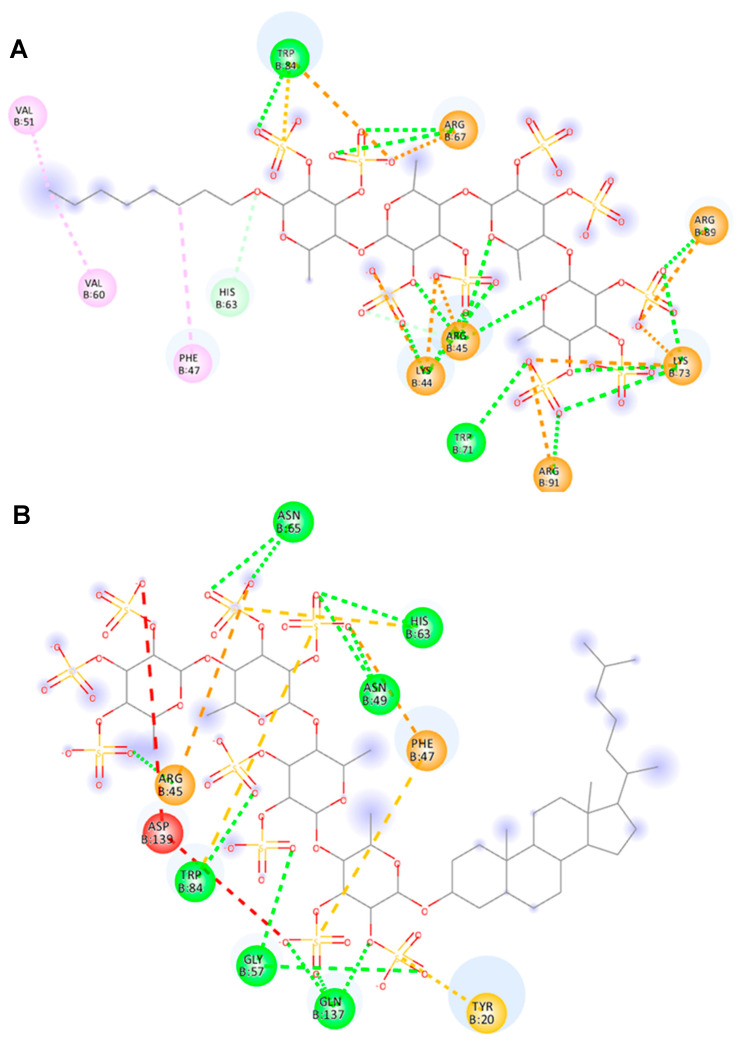
Two-dimensional interaction diagrams of docking simulation generated by BIOVIA Discovery studio (**A**) 5DUW–**10** complex, (**B**) 5DUW–**14** complex where the interacting amino acid residues are shown in circles and various interactions are demonstrated in dotted lines. Conventional hydrogen bonds (green lines), π-related interactions like π-anion (orange lines), and π-sulfur (yellow lines), salt bridge interactions (orange lines), attractive charge interactions (orange lines), alkyl and π-alkyl interactions (pink lines), unfavorable negative–negative interactions (red lines), carbon-hydrogen bond interaction (light green), respectively. 5DUW, PDB ID of human galectin-4N-CRD; PDB ID, protein data bank identifier; N-CRD, N-terminal carbohydrate-recognition domain.

**Figure 7 ijms-26-09228-f007:**
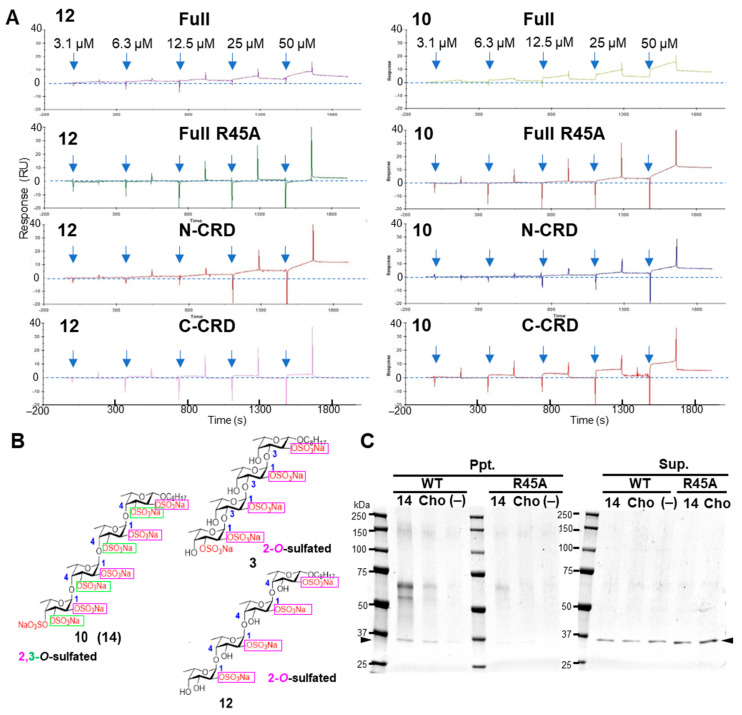
SPR and sedimentation assays for binding between fucoidan analogs and galectin-4 mutants. (**A**) The sensorgrams of SPR for binding of fucoidan analogs to galectin-4. Increasing concentrations of fucoidan analogs **12** and **10** were introduced to the full-length galectin-4 WT, mutant R45A, N-CRD and C-CRD immobilized on the surface. RU was determined by subtracting the blank values obtained on the non-immobilized surface from the values obtained on the galectin-4-immobilized surface. (**B**) Chemical structures of fucoidan analogs **1**–**4**, and **10**–**13**. Important sulfate groups are marked in pink and green. (**C**) Importance of Arg45 for the binding of galectin-4 to suspended fucoidan analog **14**. Both **14** and Cho were incubated with galectin-4 or galectin-4 R45A mutant, followed by centrifugation. The pellets (ppt.) and supernatants (sup.) were solubilized with SDS sample buffer and subjected to SDS-PAGE. The galectin-4 was visualized with Coomassie blue staining. The arrowheads indicate the position of galectin-4. Full, full-length galectin-4; R45A, R45A mutant galectin-4; N-CRD, N-terminal carbohydrate-recognition domain; C-CRD, C-terminal carbohydrate-recognition domain; WT, wild-type; Cho, cholestanol; Ppt, pellets; Sup, supernatants; (−), without.

**Figure 8 ijms-26-09228-f008:**
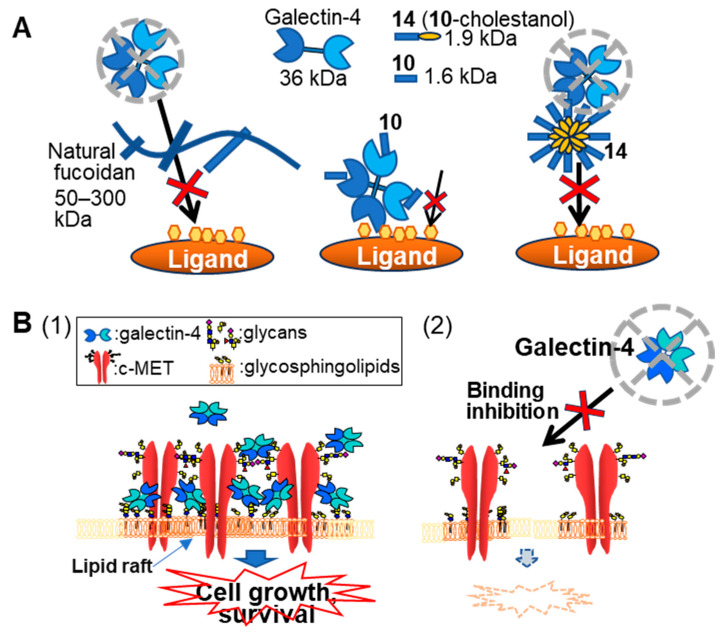
(**A**) Schematic illustration of the inhibitory effects of natural fucoidan and fucoidan analogs. Natural fucoidan with high molecular weight and fucoidan analog **14** showed high inhibitory activities. Fucoidan analogs **10** and **14** have the same glycan structure, but the multimerized **14** has increased inhibitory activity due to steric hindrance. (**B**) Schematic presentation of the proposed mechanism for galectin-4-mediated regulation. (1) In the presence of galectin-4, signaling molecules including c-MET are cross-linked and stabilized on the cell surface. (2) Galectin-4’s absence or loss of binding ability, the function of galectin-4 in signaling is considered to be weakened.

## Data Availability

The data supporting the findings of this study are available from the corresponding author upon reasonable request.

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
