# Peer review of "Inhibitory Effect of Fucoidan Analogs on Highly Metastatic Gastric Cancer Cells via Galectin-4 Inhibition"

_ijms, 2025, doi:10.3390/ijms26189228_

Round 1

Reviewer 1 Report

Comments and Suggestions for Authors

The authors in the manuscript “Inhibitory effect of fucoidan analogs on highly metastatic gastric cancer cells via galectin-4 inhibition” show that fucoidan analogs exhibited significant suppressive activity against the proliferation of malignant gastric cancer cells and propose Galectin-4 as possible mediator-pathway. The work deals with a very interesting topic, the manuscript is well written and the experiments are described in detail.

The manuscript requires some clarification and editing before it can be accepted for publication:

  • section 2.1: the authors describe the use of natural fucoidan from F. vesiculosus in the binding experiment against Gal-4, a characterization of the molecule should reported; grade of purity, moleculasr weight of the molecule, the methodology used to evalute the MW of the natural fucoidan, monodispersity value etc etc. The determination of a correct affinity costant value is dependent of a correct MW of the ligand used in the experiment!
  • Section 2.1 The authors reported SPR sensorgrams of fucoidan toward galectin-4: the dissociation of the molecule is not occurred, the molecule remain on the surface of the chip. Was the fucoidan control experiment performed on a chip without immobilized Gal-4?
  • Section 2.2: the authors present different fucoidan analogs: what is the oligomeric state of the fucoidan analogs? Are the fucoidan analogs monomers or they form oligomer?
  • Section 2.3: SPR reveals the binding behaviors of fucoidan analogs toward galectin-4: the interactions with Gal-4 of the fucoidan analogs 10 and 14 are evident but the association phase of the analog 14 is not stable, probably there are aspecific interaction, comment it.
  • The authors show that the fucoidan analogs 10 and 14 exibit suppressive activity against the proliferation of malignant gastric cancer and for treating gastric cancer metastasis, a consideration on the specifity and selectivity of the fucoidan analogs versus another galectins present in the cells analyzed should considerated.

Minor:

  • line 83-85 the authors say” Natural fucoidan …….. suppressed the phospho-c-MET (pMET)…shift here the pMET pathway desccription reported in section 3.2.
  • Figure 8. Schematic illustration of the inhibitory effects of natural fucoidan and fucoidan analogs: the authors show that the natural fucoidan binds Gal-4 and in particolar the CRD C-Term. If the authors do not have data to demonstrate this, a generic Gal-4 without domains splitting should be used in the scheme.
Comments on the Quality of English Language

The English could be improved to more clearly express the research.

Author Response

Reviewer #1:

We thank the reviewer for this valuable suggestion and for your insightful comments on our paper. We appreciate the comments, and the following are our point-by-point responses.

  • Comment 1: section 2.1: the authors describe the use of natural fucoidan from F. vesiculosus in the binding experiment against Gal-4, a characterization of the molecule should reported; grade of purity, molecular weight of the molecule, the methodology used to evaluate the MW of the natural fucoidan, monodispersity value etc etc. The determination of a correct affinity constant value is dependent of a correct MW of the ligand used in the experiment!-

Response: We thank the reviewer for this suggestion. The major problem with the natural fucoidan is its heterogeneity. Accordingly, the molecular weight of natural fucoidan is not uniform. We performed agarose electrophoresis of natural fucoidan, and a broad band was observed. Before the study on structurally distinct synthesized fucoidan analogs, we used commercial natural fucoidan in a preliminary study. Therefore, we have not carried out a precise characterization of the natural fucoidan. We found that the molecular weight of this fucoidan was stated as 464,00 Da in the "Certificate of Analysis" sheet, which was determined using the multi-angle laser light scattering method. We re-calculated the KD value assuming a molecular weight of 50 kDa. To clarify this, we added the following text in the Materials and Methods section: “We calculated the tentative dissociation constant (KD) value of natural fucoidan from F. vesiculosus, assuming a tentative molecular weight of 50 kDa.”(lines 535-537). We also described the result in line 75 as follows:” The dissociation constant (KD) value of fucoidan was tentatively determined as 2.25 × 10-8

  • Comment 2: Section 2.1 The authors reported SPR sensorgrams of fucoidan toward galectin-4: the dissociation of the molecule is not occurred, the molecule remain on the surface of the chip. Was the fucoidan control experiment performed on a chip without immobilized Gal-4?

Response: We acknowledge the reviewer’s comment and the opportunity to clarify this. The fucoidan control experiment was performed on a chip without immobilized Gal-4. This is described in figure legends of Fig.1(B), Fig.5, and Fig.7A. We added the same explanation in the Material and Methods section as follows : ”The relative response (RU) was determined by subtracting the blank values obtained on the non-immobilized surface from the values obtained on the galec-tin-4-immobilized surfaces.”(lines 532-534).

  • Comment 3: Section 2.2: the authors present different fucoidan analogs: what is the oligomeric state of the fucoidan analogs? Are the fucoidan analogs monomers or they form oligomer?

Response: From the results of SPR analysis, we speculate that the fucoidan analog 14 forms oligomer, while the others are monomers.

  • Comment 4: Section 2.3: SPR reveals the binding behaviors of fucoidan analogs toward galectin-4: the interactions with Gal-4 of the fucoidan analogs 10 and 14 are evident but the association phase of the analog 14 is not stable, probably there are aspecific interaction, comment it. -

Response: We thank the reviewer for this suggestion. As the reviewer pointed out, it is possible that the aglycone part, cholestanol, could cause unstable binding. We added the following sentence “The unstable sensorgram observed during the association phase of fucoidan analogue 14 may be due to non-specific binding of its aglycone component, cholestanol.”(lines 220-222).

  • Comment 5: The authors show that the fucoidan analogs 10 and 14 exhibit suppressive activity against the proliferation of malignant gastric cancer and for treating gastric cancer metastasis, a consideration on the specifity and selectivity of the fucoidan analogs versus another galectins present in the cells analyzed should considerated.

Response: As the reviewer pointed out, the specificity and selectivity of the fucoidan analogs versus other galectins present in the cells are important. We searched and added the expression data of galectins in NUGC4 cells (Fig. S9). And we added the following discussion. “Fucoidans may also be ligands for other galectins. As shown in Fig. S9, NUGC4 cells mainly express galectin-3 and -4. Unlike galectin-4, there is no correlation between galectin-3 expression and the peritoneal dissemination ability of gastric cancer cells [2]. Therefore, even if galectin-3 is inhibited by fucoidan, it is thought that this does not relate to the suppression of peritoneal dissemination.”(lines 407-1411).

Fig. S9.

Fig. S9. RNA expression of galectins in NUGC4 cells. RNA expression data as normalized transcript per million (nTPM) values of NUGC4 cells are graphed. LGALS5, LGALS6, LGALS5, LGALS10, and LGALS11 are expressed in other species. We obtained nTPM data of human galectin genes from the human protein atlas (HPA) database (https://www.proteinatlas.org/ (accessed on 12 August 2025))[45].

Minor:

  • Comment 6: line 83-85 the authors say” Natural fucoidan …….. suppressed the phospho-c-MET (pMET)…shift here the pMET pathway description reported in section 3.2.

Response: Following the reviewer’s suggestion, we shifted the pMET pathway description in section 3.2. to section 2.1.  phospho-c-MET (pMET), caused by autophosphorylation of c-MET [2] (line 84).

  • Comment 7: Figure 8. Schematic illustration of the inhibitory effects of natural fucoidan and fucoidan analogs: the authors show that the natural fucoidan binds Gal-4 and in particular the CRD C-Term. If the authors do not have data to demonstrate this, a generic Gal-4 without domains splitting should be used in the scheme.

Response: We thank the reviewer for this suggestion. As the reviewer pointed out, we do not have data that shows the natural fucoidan specifically binds Gal-4 C-CRD. Accordingly, we revised the Fig.8A as follows to reflect this.

Reviewer 2 Report

Comments and Suggestions for Authors

Your research is excellent and promising.

Author Response

Reviewer 2

Comments and Suggestions for Authors

  • Comment Your research is excellent and promising.

Response: Thank you very much for your comment.

Reviewer 3 Report

Comments and Suggestions for Authors

The manuscript "Inhibitory effect of fucoidan analogs on highly metastatic gastric cancer cells via galectin-4 inhibition" addresses the important clinical problem of peritoneal dissemination of gastric cancer, which remains largely untreatable and novel molecular strategies are needed. The work explores a mechanistic link between structurally defined fucoidan analogs and galectin-4–mediated signaling, which has not been previously demonstrated with this level of molecular detail. The novelty lies in the systematic evaluation of synthetic fucoidan analogs with defined sulfation and aglycone structures. The study is suitable for publication in IJMS but needs major revision before it can be accepted. My decision is based on the comments, presented below, organized by manuscript sections.

The abstract needs to be restructured according to IJMS guidelines for authors into four clear parts: background, methods, results, and conclusion. Include sub-sections or clear transitions for each part for easy readability.

In the abstract, present the clinical context moving in the direction from the general to specific concepts. For example, first introduce gastric cancer, then peritoneal dissemination and finally galectin-4.

Briefly explain the rationale for the selected fucoidan analogs and what are the distinct favorable characteristics of the tested structures.

Mention the main methodological approaches in the abstract. Currently, only endpoints like “proliferation” and “mutagenesis” are mentioned, which seems out of context without the methods.

In the beginning of the introduction clarify that peritoneal dissemination is a metastatic process within the peritoneal cavity rather than a separate cancer type.

Expand the description of fucoidan to include current knowledge about its therapeutic potential, pharmacokinetics, possible application routes (i.v., oral, etc.), tissue distribution and inherent limitations.

Conclude the introduction with a clear study aim derived from knowledge gaps rather than repeating the conclusions.

In the results section, adhere to IJMS guidelines for authors and remove discussion content, keeping only direct data interpretation.

Use a single, consistent unit of concentration throughout the text, tables, and figures. For example, convert all μg/mL and μM units to one single unit and apply it uniformly in text and plots.

Report the ratio of association and dissociation constants in the SPR data to facilitate interpretation of binding strength.

Introduce all abbreviations, such as c-MET and p-MET at first mention and briefly describe their biological roles.

Rephrase the following sentence on c-MET expression and post-translational effects: “The relative expression level of c-MET did not change in treated cells, suggesting a post-translational effect, whereas decreased galectin-4 expression suppressed c-MET and p-MET [2].” so that causality is clearly understood.

Perform standard redocking with the co-crystallized ligand, report RMSD values, and compare docking scores with the native ligand. Include this as a reference baseline in plots for test compounds.

Define explicit criteria for selecting the optimal docking poses and compare each compound’s interactions to the native ligand. For example, report which bonds mimic the native ligand and highlight which of the studied compounds is most similar in orientation and interactions.

In figures, harmonize the concentration units to match the unified system, chosen in the text.

Revise figure captions to describe the method and endpoint measured without interpreting results. For instance, specify “ATP-based cell viability assay” rather than stating “cells were inhibited.”

Replace generic y-axis labels like “response” with specific endpoints and their units.

Explain all abbreviations in figure captions, for example “Cho” in Fig. 3.

Improve the resolution and clarity of Figure 6 to make the molecular interactions and labels easily legible.

Introduce all compound structures together in Figure 2 and remove duplicated molecular structure content in Figure 7.

Move Figure 8 in the Discussion section immediately after the description of the proposed mechanism. Currently, it is placed after the conclusion where it is disconnected.

In the Discussion section, clearly explain the difference between both tested compound groups and why group B includes only one cholestanol conjugate compound. Explain the basis for its selection and which of the unconjugated analogs in group A its structure resembles the most.

Discuss why compounds 3 and 10 were not conjugated to cholestanol and what effect conjugation would likely have on their activity.

Address the presence of non-galectin-specific cytotoxic effects and discuss likely mechanisms, including in the context of the apoptosis assay results, stating that caspase-3 activation was not observed.

Discuss the inactivation of galectin-4 activity in the presence of serum and the implications for intravenous application in vivo.

Explain how docking interactions relate to the redocked native ligand in the context of suggested structural optimizations to improve affinity. For example, indicate if adding sulfates or modifying the aglycone would likely improve binding.

In the Materials and methods section, except for the provided reference, describe the main steps for preparing galectin-4, the R45A mutant, and the N- and C-terminal domains. Explain the logic and key steps of the ELISA, SPR, ATP assay, and sedimentation experiments to ensure reproducibility.

Include a description of the redocking validation procedure in the methods, including RMSD comparison with the native ligand poses.

Comments on the Quality of English Language

Revise phrasing to use precise prepositions such as “inhibitory activity of galectin-4” instead of “inhibitory activity towards galectin-4.”

Author Response

Reviewer #3:

Thank you very much for your insightful comments on our paper. We appreciate the comments, and the following are our point-by-point responses.

  • Comment 1: The abstract needs to be restructured according to IJMS guidelines for authors into four clear parts: background, methods, results, and conclusion. Include sub-sections or clear transitions for each part for easy readability.

Response: We thank the reviewer for this valuable suggestion. As the reviewer suggested, we rewrote the abstract following the style of structured abstracts, but without headings. Because the editorial office emailed us as follows, “This does not follow our journal guidelines. The abstract should not contain the heading titles mentioned by the Reviewer. Please do not include such a change during your revision.”

  • Comment 2: In the abstract, present the clinical context moving in the direction from the general to specific concepts. For example, first introduce gastric cancer, then peritoneal dissemination and finally galectin-4.

Response: Following the reviewer’s advice, we changed the abstract as follows. “In malignant type gastric cancer, peritoneal dissemination is the most frequent metastatic process and is an inoperable condition for which effective treatment is lacking. Our research has revealed that galectin-4 plays an important role in the peritoneal metastasis of gastric cancer cells.” (lines 10-13)

  • Comment 3: Briefly explain the rationale for the selected fucoidan analogs and what are the distinct favorable characteristics of the tested structures.

Response: Following the reviewer’s comment, we explain the selected fucoidan analogs and what are the distinct favorable characteristics of the tested structures as follows. “Among the 13 fucoidan analogs tested, analog 10, whose sugar chains composed of repeating 2,3-O-sulfated α(1,4)-linked L-fucose, showed significant inhibitory activity against galectin-4 binding and cell proliferation.” (lines 19-22)

  • Comment 4: Mention the main methodological approaches in the abstract. Currently, only endpoints like “proliferation” and “mutagenesis” are mentioned, which seems out of context without the methods. -

Response: Following the reviewer’s advice, we mentioned the main methodological approaches in the abstract. “The inhibitory activity towards galectin-4 binding was evaluated using a competitive ELISA method, while the suppressive effect on gastric cancer cell proliferation was assessed using an ATP-based cell viability assay. Direct binding to galectin-4 was examined by surface plasmon resonance analysis.” (lines 14-17)

  • Comment 5: In the beginning of the introduction clarify that peritoneal dissemination is a metastatic process within the peritoneal cavity rather than a separate cancer type

Response: Following the reviewer’s advice, we added the sentence as follows.” In malignant type gastric cancer, peritoneal dissemination is a most frequent metastatic process within the peritoneal cavity.” (lines 31-32)

  • Comment 6: Expand the description of fucoidan to include current knowledge about its therapeutic potential, pharmacokinetics, possible application routes (i.v., oral, etc.), tissue distribution and inherent limitations.

Response: We acknowledge the reviewer’s comment. However, we are sorry to say that current knowledge about the therapeutic potential and pharmacokinetics of fucoidan is difficult to explain, because many studies examine different types of fucoidan on different cell lines or animal models. In addition, there is relatively little information available with regard to the absorption, distribution, and excretion of fucoidan. Our research focused mainly on fucoidan analogs with distinct structures, and we studied natural fucoidan as an introduction to this. Adding detailed explanations about natural fucoidan, as suggested by the reviewer, may blur the focus of this study. We already included the description of the current knowledge of the anti-cancer effect of fucoidans with relevant literature references (lines 51-53) in the manuscript. Based on the reviewer comment, we have added several references [8-13,15] and incorporated the pharmacodynamics of fucoidan additionally (lines 53-54). The added text is highlighted as follows.: The anticancer properties of natural fucoidans have been reported in both in vivo and in vitro studies for various types of cancers [11,12]. However, the studies on the pharmacodynamics of fucoidan remain relatively limited [13] (lines 51-54).  

  • Comment 7: Conclude the introduction with a clear study aim derived from knowledge gaps rather than repeating the conclusions.

Response: We thank the reviewer for this suggestion. Following the reviewer’s advice, we conclude the introduction with a study aim and deleted the conclusion part as follows. Deleted the following sentence: “Using the synthesized fucoidan analogs with well-defined structures -- -- -- gastric cancer cells and their binding ability to galectin-4. In this study, we found that natural fucoidan ---- --- -- enzyme-linked immunosorbent assay (ELISA).

Instead, we added the following sentence: ” This study aims to clarify the structure-activity relationship in the suppression of peritoneal metastasis through systematic evaluation of synthetic fucoidan analogs with well-defined structures.” (lines 62-64)

  • Comment 8: In the results section, adhere to IJMS guidelines for authors and remove discussion content, keeping only direct data interpretation. -  

Response: Thank you for your suggestion. However, we believe that it would be easier to understand if we presented a discussion on each result alongside the results themselves. This style does not contravene the journal's guidelines. Because the “Discussion section” of the IJMS “Instructions for Authors Manuscript Preparation” explains that “This section may be combined with Results.”.

  • Comment 9: Use a single, consistent unit of concentration throughout the text, tables, and figures. For example, convert all μg/mL and μM units to one single unit and apply it uniformly in text and plots. -

Response: We agree that we should use a single, consistent unit throughout the text. However, as the molecular weight of natural fucoidan is not uniform, it is difficult to express its molar concentration accurately. Furthermore, as far as we have investigated, most papers expressed the concentration of natural fucoidan in μg/ml or mg/ml. On the other hand, μM units are appropriate for comparing the inhibitory activity of low-molecular-weight fucoidan analogs. So, we use μg/ml for natural fucoidan, and μM for fucoidan analogs.

  • Comment 10: Report the ratio of association and dissociation constants in the SPR data to facilitate interpretation of binding strength.

Response: We thank the reviewer for this suggestion. However, we are sorry to say that an accurate ratio of association and dissociation constants in the SPR data could not be determined, because it was difficult to apply high concentrations of fucoidan analogs owing to solubility concerns.

  • Comment 11: Introduce all abbreviations, such as c-MET and p-MET at first mention and briefly describe their biological roles.

Response: We thank the reviewer for this suggestion. We described abbreviations and mentioned their role briefly as follows : ”We observed that suppression of galectin-4 expression inhibited the expression of c-MET, hepatocyte growth factor receptor that has tyrosine kinase activity and phospho-c-MET (pMET), caused by autophosphorylation of c-MET [2].” (lines 82-84)

  • Comment 12: Rephrase the following sentence on c-MET expression and post-translational effects: “The relative expression level of c-MET did not change in treated cells, suggesting a post-translational effect, whereas decreased galectin-4 expression suppressed c-MET and p-MET [2].” so that causality is clearly understood.

Response: Following the reviewer’s advice, we rephrased the sentence as follows: “The relative genetic expression level of c-MET did not change in treated cells, suggesting that the reduction occurred by post-translational effect (data not shown). These results are consistent with the finding that the reduction of c-MET and pMET occurred by the suppression of galectin-4 [2].” (lines 186-190)

  • Comment 13: Perform standard redocking with the co-crystallized ligand, report RMSD values, and compare docking scores with the native ligand. Include this as a reference baseline in plots for test compounds.

Response: We thank the reviewer for this valuable suggestion and acknowledge the importance of redocking as a validation step. In our study, the target protein, galectin-4N-terminal carbohydrate recognition domain (Gal-4N CRD, PDB ID: 5DUW), is co-crystallized with lactose-3′-sulfate (3-O-sulfo-β-D-galactopyranose-(1-4)-β- D-glucopyranose), which is also its native ligand. Since both are the same molecule, a comparative docking score analysis is not possible, as it would inevitably yield identical results.

In response to the reviewer’s comment, we carried out redocking of lactose-3′-sulfate into the 5DUW binding site using the previously reported protocol (ref. 26). The redocking successfully reproduced the crystallographic pose (RMSD 0.0 Å) with a binding energy of −7.0 kcal·mol⁻¹. The detailed parameters are provided in Section 4.10 (Materials and Methods, lines 558-574), and the results are described in Section 2.5 (Results and Discussion, lines 309-317). An additional figure has been included in the revised Supporting Information (Fig. S8), where the co-crystallized ligand and the redocked ligand (pink) are shown for clarity.

Fig. S8. Redocking of lactose-3′-sulfate into the binding site of 5DUW. The crystal structure of the monomeric chain of 5DUW is shown as a ribbon diagram (β-strands in cyan, α-helices in red, and loops in grey/green). The co-crystallized native ligand, lactose-3′-sulfate, is shown in its original position, while the redocked pose is highlighted in pink.

  • Comment 14: Define explicit criteria for selecting the optimal docking poses and compare each compound’s interactions to the native ligand. For example, report which bonds mimic the native ligand and highlight which of the studied compounds is most similar in orientation and interactions.

Response: We acknowledge the reviewer’s comment and the opportunity to clarify our rationale. In line with the suggestion, we have added the explicit criteria and detailed docking protocols in the revised manuscript (Materials and Methods, Section 4.10), as shown below.

“The crystal structure of 5DUW is a tetramer comprising four identical chains, each co-crystallized with the ligand lactose-3′-sulfate, which also serves as the native ligand. For docking studies, a single chain containing the CRD was retained, while the remaining chains were removed. Protein preparation included the removal of non-essential water molecules, ions, and heteroatoms, the addition of polar hydrogens and charges, and modeling of missing residues to ensure a complete and stable structure. The chemical structures of compounds 10 and 14 were drawn in ChemDraw and energy-minimized using OpenBabel. Molecular docking was performed with Auto-Dock Vina embedded in PyRx, using a 50 × 50 × 50 Å grid box centered to encompass all key active site residues [43]. For each ligand, 20 poses were generated, ranked according to binding affinity, and visually inspected. The top docking poses were selected based on lowest binding energies and preservation of critical interactions, and the resulting complexes were further analyzed with BIOVIA discovery studio for 2D and PyMOL for 3D visualization [44]. To validate the docking protocol, the co-crystallized ligand, lactose-3′-sulfate, was redocked into the binding site of 5DUW using the same parameters applied to our test compounds; however, a smaller grid box (25 × 25 × 25 Å) was used to accommodate the reduced size of the ligand and the active site.” (lines 556-572)

We would like to clarify that our study was not designed to compare the docking interactions of compounds 10 and 14 directly with those of the native ligand. We did not aim to imply that our compounds exhibit superior or inferior binding relative to lactose-3′-sulfate. Rather, the purpose of the docking study was to provide a structural rationale for the higher growth inhibitory activity observed for 14 compared to 10, with particular emphasis on the role of Arg45. It should also be noted that 10 and 14 are low-molecular-weight heparin mimics with substantially larger size and molecular weight than the native ligand. Consequently, a larger docking grid was required to accommodate their dimensions while still encompassing all relevant active residues (as described in ref. 26). Given these structural differences, a one-to-one bond mimicry with the native ligand is not scientifically justified. Nonetheless, careful inspection of the docking poses confirmed that both ligands retained key contacts within the binding site, thereby supporting the biological relevance of the predicted binding modes.

  • Comment 15: In figures, harmonize the concentration units to match the unified system, chosen in the text.

Response: We thank the reviewer for this suggestion. However, as we explained in our response to comment 9, we use μg/ml for natural fucoidan and μM for fucoidan analogs.

  • Comment 16: Revise figure captions to describe the method and endpoint measured without interpreting results. For instance, specify “ATP-based cell viability assay” rather than stating “cells were inhibited.”

Response: Following the reviewer’s advice, we changed the figure caption in Fig.1 as follows.

  • Fucoidan inhibited the binding -----“ à “ELISA-based inhibition assay of fucoidan for the binding ----”
  • Fucoidan inhibited the proliferation of MKN45 cells. à” ATP-based cell viability assay of fucoidan on MKN45 cells.
  • Protein expression levels of pMET, c-MET, and β-actin were determined by western blotting. àThe western blot analysis of fucoidan treated MKN45 cells with pMET, c-MET, and β-actin antibodies.”
  • Fucoidan inhibited the proliferation of NUGC4 WT and KO cells. à “ATP-based cell viability assay of fucoidan on NUGC4 WT and KO cells.”

  • Comment 17: Replace generic y-axis labels like “response” with specific endpoints and their units.

Response: Thank you for your suggestion; however, response for y-axis in SPR experiments are commonly used. We have never seen any other labels for the y-axis of SPR.

  • Comment 18: Explain all abbreviations in figure captions, for example “Cho” in Fig. 3. -

Response: We acknowledge the reviewer’s comment and we apologize for the omission of the explanation. We explained all abbreviations in the figure captions of the revised manuscript.   

  • Comment 19: Improve the resolution and clarity of Figure 6 to make the molecular interactions and labels easily legible.

Response: We thank the reviewer for this helpful suggestion. Figure 6 has been regenerated at higher resolution to enhance legibility of molecular interactions and labels. The updated figure is included in the revised manuscript.

  • Comment 20:Introduce all compound structures together in Figure 2 and remove duplicated molecular structure content in Figure 7.

Response: We thank the reviewer for this suggestion. Following the reviewer’s advice, we changed the Fig. 7B to explain the favorable characteristics of the structure clearly. The updated figure is included in the revised manuscript.

  • Comment 21: Move Figure 8 in the Discussion section immediately after the description of the proposed mechanism. Currently, it is placed after the conclusion, where it is disconnected.

Response: Following the reviewer’s advice, we moved Fig. 8 immediately after the description of the proposed mechanism.

  • Comment 22: In the Discussion section, clearly explain the difference between both tested compound groups and why group B includes only one cholestanol conjugate compound. Explain the basis for its selection and which of the unconjugated analogs in group A its structure resembles the most.

Response: We apologize for any misunderstanding caused by our insufficient explanation. The structure of fucoidan is important for affinity, not the aglycon part cholestanol. Our strategy was to first select the most powerful fucoidan analog from Group A. Our recent study revealed that 10 exhibits the strongest inhibitory activity against SARS-CoV-2 [17], which has binding activity similar to galectin-4 [32]. Fucoidan analogue 14, in which the octyl group of 10 was changed to a cholestanyl group, was found to show enhanced inhibitory activity against SARS-CoV-2. With this research background, we investigated whether a similar phenomenon occurs with galectin-4. This discussion is described in lines 384-392.  We also added the following explanation for 14 (cholestanyl group as the aglycone moiety instead of the octyl group in 10) into the figure caption of Fig.2.

  • Comment 23: Discuss why compounds 3 and 10 were not conjugated to cholestanol and what effect conjugation would likely have on their activity.

Response: We apologize for any misunderstanding caused by our insufficient explanation. We added the following explanation for 14 (cholestanyl group as the aglycone moiety instead of the octyl group in 10) into the figure caption of Fig.2.

  • Comment 24: Address the presence of non-galectin-specific cytotoxic effects and discuss likely mechanisms, including in the context of the apoptosis assay results, stating that caspase-3 activation was not observed.

Response: We thank the reviewer for the suggestion about the presence of non-galectin-specific cytotoxic effects. Following the reviewer’s comment, we added the following text in the general discussion. “There are many reports that natural fucoidan activates caspase-3. Since no activation was observed at the concentrations tested with fucoidan analogs (Figs. 4C and S4), suggesting that caspase-3 activation occurs via a pathway that does not involve galectin-4 (or galectin-4 involves it to a lesser extent). It is also possible that the activation of caspase-3 induced by galectin-4 remain below the detection limit. In any case, since growth suppression was not observed at the tested concentrations in HEK293 and NUGC4 KO cells, which lack galectin-4 expression, indicating that the non-galectin-4-specific cytotoxic effects of fucoidan analogs are low.” (lines 365-372)

  • Comment 25: Discuss the inactivation of galectin-4 activity in the presence of serum and the implications for intravenous application in vivo.

Response: We thank the reviewer for this suggestion. The cause of galectin-4 inactivation in serum is currently unknown. Peritoneal dissemination occurs when cells detach into the abdominal cavity rather than into blood vessels. For this reason, we did not consider this discussion to be necessary.

  • Comment 26: Explain how docking interactions relate to the redocked native ligand in the context of suggested structural optimizations to improve affinity. For example, indicate if adding sulfates or modifying the aglycone would likely improve binding.

Response: Our docking study aimed to provide a theoretical rationale for the higher growth-inhibitory activity observed with the low-molecular-weight heparin (LMWH) mimic bearing a hydrophobic cholestanyl aglycone 14, as compared to its structural analogue carrying an n-octyl aglycone 10. Since both compounds share the same glycone backbone and differ only in their aglycone moiety, their docking energies and molecular interactions could be directly compared, allowing us to attribute differences in binding to the presence of the sterol tail. In contrast, a similar comparison with the native ligand, lactose-3′-sulfate, is scientifically inappropriate because it is a much smaller disaccharide with significantly different molecular weight, topology, and chemical properties. While one could in principle, compare the role of sulfate groups between the test compounds and the native ligand, such an analysis lies beyond the scope of the present study. In line with the reviewer’s comment, we noted that 14 exhibited a slightly higher binding energy (–7.1 kcal·mol⁻¹) than the native ligand (–7.0 kcal·mol⁻¹), while 10 showed a lower value (–6.1 kcal·mol⁻¹). These small numerical differences, however, should not be over-interpreted, as they largely arise from the inherent structural disparities between the small disaccharide native ligand and the bulkier LMWH mimics. It is important to specify here that the intention of our study was not to compare of the binding interactions of our tested compounds with those of the native ligand.

  • Comment 27: In the Materials and methods section, except for the provided reference, describe the main steps for preparing galectin-4, the R45A mutant, and the N- and C-terminal domains. Explain the logic and key steps of the ELISA, SPR, ATP assay, and sedimentation experiments to ensure reproducibility. -

Response:In accordance with the reviewer's comments, we added more detailed explanations in the Materials and Methods section. The added detailed explanations are highlighted in the revised manuscript. (lines 457-471, 514-525)

  • Comment 28: Include a description of the redocking validation procedure in the methods, including RMSD comparison with the native ligand poses.

Response: We thank the reviewer for this suggestion. A description of the redocking validation procedure, including RMSD comparisons with the native ligand poses, has been added to the Materials and Methods section (section. 4.10) to provide a clear reference baseline for our docking study.

Comments on the Quality of English Language

  • Comment 29: Revise phrasing to use precise prepositions such as “inhibitory activity of galectin-4” instead of “inhibitory activity towards galectin-4.”

Response: Following the reviewer’s comment, we replaced “inhibitory activity of galectin-4” to “inhibitory activity towards galectin-4.” (line 48)

Round 2

Reviewer 1 Report

Comments and Suggestions for Authors

The manuscript has been improved and in its present form can be accepted for publication.

Author Response

Reviewer 1

Comments and Suggestions for Authors

  • Comment : The manuscript has been improved and in its present form can be accepted for publication.

Response: Thank you very much for your comment.

Reviewer 3 Report

Comments and Suggestions for Authors

While the authors have addressed many of my previous comments, there are points that warrant further clarification.

Provide a more detailed discussion regarding the implications of galectin-4 inactivation by serum components and how this may affect potential intravenous or other possible routes of administration and in vivo applicability. Although such experimental studies are lacking, implement in the discussion some general principles for the relationships between molecular mass and surface charge and ADME properties.

Although the rationale for using different concentration units for natural fucoidan and synthetic analogs was explained, consider ways to harmonize or clearly cross-reference these measures in the text and figures for enhanced clarity. Include a concise summary table comparing the binding affinities, inhibitory concentrations, and structural features of all tested fucoidan analogs to make structure-activity relationships more understandable.

The discussion describing why compounds 3 and 10 were not conjugated to cholestanol and the predicted impact of such conjugation on their activity remains limited and should be expanded.

Discuss more explicitly the observed non-galectin-4 specific cytotoxic effects, integrating the apoptosis assay results to clarify possible alternative pathways involved.

The authors report an RMSD of 0.0 Å for their redocking of the co-crystallized ligand lactose-3′-sulfate to the galectin-4 N-terminal carbohydrate recognition domain. Such an exact RMSD is highly unusual in molecular docking studies and may indicate potential issues with the docking or analysis process. Typically, successful redocking yields RMSD values in the range of 0.5–1.5 Å due to minor variability in ligand conformations and protein flexibility. Therefore, I recommend that the authors carefully verify and clarify their docking protocol and RMSD calculation methodology to ensure that the ligand was actively re-docked rather than simply reloaded or aligned without sampling. Visual confirmation of the docking poses and a detailed explanation of the RMSD computation would strengthen confidence in the docking validation.

Provide a clearer comparative commentary on how the obtained docking results inform further structural optimization strategies, such as specific sulfation patterns or aglycone modifications would improve affinity.

I recommend that the manuscript be published only after all of the above points are comprehensively addressed.

Author Response

We acknowledge the reviewer’s comments and the opportunity to see our study from a new perspective. We appreciate the comments, and the following are our point-by-point responses.

Comment 1: Provide a more detailed discussion regarding the implications of galectin-4 inactivation by serum components and how this may affect potential intravenous or other possible routes of administration and in vivo applicability. Although such experimental studies are lacking, implement in the discussion some general principles for the relationships between molecular mass and surface charge and ADME properties.

Response: We appreciate the reviewer's suggestion. In line with the suggestion, we have added the discussion as follows: “Characterization of the absorption, distribution, metabolism, and excretion (ADME) properties is important for drug administration. However, such experimental studies are lacking for galectin-4. Generally, surface charge mediates interactions with various cellular and extracellular components, while mass affects diffusion and clearance mechanisms [38]. The molecular weight of galectin-4 is approximately 36 kDa, such a smaller protein has faster tissue penetration but more rapid elimination, leading to a shorter duration in the circulation. Furthermore, galectin-4 may bind to specific glycoproteins or glycolipids in serum; this is inferred from the fact that both galectin-4 binding and growth inhibitory activities of natural fucoidan were attenuated in the presence of serum. Structural study revealed that positively charged residues are located on the surface of galectin-4 [39]. Galectin-4 is expressed mainly in the epithelial cells of the gastrointestinal tract, and its surface charge facilitates its interactions with cells and tissues, forming complexes with other membrane proteins and lipids. Metabolism of galectin-4 is influenced by interactions with cellular components, such as lipid rafts, and is part of its functional role. Considering all of this background, intraperitoneal administration may be preferable to intravenous administration.” (lines 420-435)

Comment 2: Although the rationale for using different concentration units for natural fucoidan and synthetic analogs was explained, consider ways to harmonize or clearly cross-reference these measures in the text and figures for enhanced clarity. Include a concise summary table comparing the binding affinities, inhibitory concentrations, and structural features of all tested fucoidan analogs to make structure-activity relationships more understandable.

Response: Following the reviewer's advice, we considered the way to cross-reference concentration units for natural fucoidan and synthetic analogs. As we previously explained, the molecular weight of natural fucoidan is not uniform, and presenting the horizontal axis as molar concentration would lack accuracy. And most papers expressed the concentration of natural fucoidan in μg/ml or mg/ml. To enable readers to easily calculate molar concentration, we added the following text in the Figure legends.

“Assuming the molecular weight of natural fucoidan is 50 kDa, 100 μg/ml is equivalent to 2 μM.”

(lines 115-116)

Comment 2, latter half: Include a concise summary table comparing the binding affinities, inhibitory concentrations, and structural features of all tested fucoidan analogs to make structure-activity relationships more understandable.

Response: Following the reviewer’s advice, we included a concise summary table comparing the fucose linkage, sulfated pattern, growth inhibitory, and galectin-4 binding inhibitory concentrations of all tested fucoidan analogs to make structure-activity relationships more understandable.

(Supplementary information Table S1)

Comment 3: The discussion describing why compounds 3 and 10 were not conjugated to cholestanol and the predicted impact of such conjugation on their activity remains limited and should be expanded.

Response: We apologize for our insufficient explanation. Our strategy was to first select the most powerful fucoidan analog from fucoidan analogs 1~13. Our recent study revealed that 10 exhibits the strongest inhibitory activity against SARS-CoV-2 [17], which has binding activity similar to galectin-4 [33]. Fucoidan analogue 14, cholestanol-conjugated form of 10 was found to show enhanced inhibitory activity against SARS-CoV-2 [17]. We had planned to conjugate with cholestanol if any of fucoidan analogs 1~13 proved more promising than 10. However, none of the fucoidan analogs (including 3) surpassed 10, particularly concerning proliferation-inhibiting activity. Since the fucoidan analog 14 is the cholestanol-conjugated form of 10, comparing 10 and 14 allows us to explain the effect of cholestanol-conjugation.

To reflect the above points, we have rewritten the manuscript as follows.

“Our strategy was to first select the most potent fucoidan analog from fucoidan analogs 1~13 [17]. The fucoidan analog 14 [18] (Fig. 2B), comprising the cholestanyl group as the aglycone moiety instead of the octyl group in 10 was also studied, because cholestanol-conjugation could potentially increase inhibitory activity.” (lines 123-127)

“Our strategy was to first select the most potent one from fucoidan analogs 1~13. We had planned to conjugate with cholestanol if any of fucoidan analogs 1~13 proved more promising than 10. However, none of the fucoidan analogs (including 3) surpassed 10, particularly concerning proliferation-inhibiting activity (Table S1). (lines 394-398)

The predicted impact of cholestanol-conjugation can be evaluated by comparing the activities of fucoidan analogs 10 and 14 (cholestanol-conjugated form of 10). Chemically synthesized sugar cholestanol inhibits the proliferation of colorectal and gastric cancer cells [34]. Although there are differences between their study and ours in the method of administration to cells, they showed that the cholestanol-conjugated compounds were rapidly taken up via lipid rafts/microdomains on the cell surface and altered the expression levels of apoptosis-related molecules in cancer cells. High-molecular-weight natural fucoidan exhibits high inhibitory activity even when part of its structure binds to galectin-4. It is interesting to mention that 10 and 14 possess the same glycone moiety that binds to galectin-4. However, 14 exhibited higher growth inhibitory and galectin-4 binding inhibitory activities than 10 did. The large SPR response suggests multimerization of 14, which may also lead to a significant increase in the inhibitory activity due to steric inhibition (Fig. 8A). The weak binding interaction between lectins and carbohydrate moieties can be compensated by the multivalent display of glycans [35,36]. In particular, multivalent ligands substantially enhance binding to galectin-4 [37]. Collectively, conjugation with cholestanol may synergistically enhance the anticancer activity of fucoidan analogs. (lines 399-414)

Comment 4: Discuss more explicitly the observed non-galectin-4 specific cytotoxic effects, integrating the apoptosis assay results to clarify possible alternative pathways involved.

Response: Thank you for your suggestion. We apologize for the lack of clarity in our manuscript. We rewrote the discussion as follows: “Non-galectin-4 specific cytotoxic effects can be investigated by using cells that lack galectin-4 expression. Growth suppression was observed in NUGC4 KO cells, which lack galectin-4 expression, with natural fucoidan and fucoidan analog 14, suggesting non-galectin-4-specific cytotoxic effects. Natural fucoidans induce apoptosis in various cancer cells through both the intrinsic mitochondrial pathway and the extrinsic death receptor pathway, often involving the activation of caspases, altered Bcl-2/Bax ratios, and the modulation of MAPK/ERK and PI3K/Akt signaling pathways [12]. Fucoidan analog 14 did not activate caspase-3 at the low concentrations tested. However, at high concentrations, the apoptosis pathway reported for natural fucoidan may be activated, necessitating further investigation.” (lines 368-377)

Comment 5: The authors report an RMSD of 0.0 Å for their redocking of the co-crystallized ligand lactose-3′-sulfate to the galectin-4 N-terminal carbohydrate recognition domain. Such an exact RMSD is highly unusual in molecular docking studies and may indicate potential issues with the docking or analysis process. Typically, successful redocking yields RMSD values in the range of 0.5–1.5 Å due to minor variability in ligand conformations and protein flexibility. Therefore, I recommend that the authors carefully verify and clarify their docking protocol and RMSD calculation methodology to ensure that the ligand was actively re-docked rather than simply reloaded or aligned without sampling. Visual confirmation of the docking poses and a detailed explanation of the RMSD computation would strengthen confidence in the docking validation.

Response: There is a misunderstanding between us and Reviewer. Our theoretical calculations involve only molecular docking with AutoDock Vina rather than Molecular Dynamics. The co-crystallized ligand (lactose-3′-sulfate) was fully removed from the binding site prior to docking, and the best-ranked pose reproduced the crystallographic coordinates with an RMSD of 0.0 Å. Such exact overlap is possible when the ligand is relatively rigid and the binding pocket is well-defined. The Supporting Information (Fig. S8) shows the re-docked pose (pink), confirming that the ligand was actively re-docked rather than simply reloaded or aligned without sampling.

Comment 6: Provide a clearer comparative commentary on how the obtained docking results inform further structural optimization strategies, such as specific sulfation patterns or aglycone modifications would improve affinity.

Response: The redocking strategy enabled us to systematically assess the critical contribution of sulfate residues in mediating interactions with amino acids within the galectin-4 N-terminal carbohydrate recognition domain. Our analysis demonstrated that the negative charge of the sulfate groups is essential for establishing strong electrostatic interactions and salt bridges with positively charged residues such as Arginine (ARG) and Lysine (LYS) Building on these insights, we designed ligand candidate 10 with an extended sugar chain containing additional sulfate groups, aiming to enhance these favorable interactions. Subsequent docking simulations of 10 confirmed our hypothesis, revealing not only an additive effect from the extra sulfate groups but also significant contributions from hydrophobic interactions that help stabilize the ligand within the binding pocket. To further explore and validate the role of hydrophobic contacts, we modified the aglycone moiety by introducing a cholestanyl group in ligand 14, which reinforced the importance of hydrophobic interactions in ligand binding. Collectively, these findings provide a rational framework for designing galectin-4N ligands with optimized sulfation patterns and tailored aglycone modifications to maximize binding affinity.

Round 3

Reviewer 3 Report

Comments and Suggestions for Authors

Dear Authors,

Most of my previous comments have been adequately addressed, particularly those concerning the discussion of ADME properties, apoptosis assay results, and the commentary on structural optimization. However, I still encourage the authors to provide clearer cross-referencing of concentration units in the text beyond the single note in the figure legends. However, the RMSD calculation method should be clearly described in the Methods section to strengthen confidence in the docking validation. Figure S8 shows that the re-docked lactose-3′-sulfate overlays closely with the crystallographic pose, but not perfectly. The two poses display small, visible deviations and the RMSD of 0.0 Å cannot be correct for this comparison.

Author Response

Reviewer #3:

We thank the reviewer for the suggestion on our paper. We appreciate the comments, and the following are our point-by-point responses.

  • Comment 1: Most of my previous comments have been adequately addressed, particularly those concerning the discussion of ADME properties, apoptosis assay results, and the commentary on structural optimization. However, I still encourage the authors to provide clearer cross-referencing of concentration units in the text beyond the single note in the figure legends.

Response: Following the reviewer's advice, we added the following text in the main text as follows: “Although fucoidan has a heterogeneous molecular weight, assuming it to be 50 kDa, 3 μg/ml is equivalent to 0.06 μM.” (lines 72-74)

  • Comment 2: However, the RMSD calculation method should be clearly described in the Methods section to strengthen confidence in the docking validation. Figure S8 shows that the re-docked lactose-3′-sulfate overlays closely with the crystallographic pose, but not perfectly. The two poses display small, visible deviations and the RMSD of 0.0 Å cannot be correct for this comparison.

Response: As per the reviewer’s suggestion, we have incorporated a detailed description of the RMSD calculation procedure in the Materials and Methods section of the revised manuscript. The added text reads as follows: “RMSD calculations were performed using Discovery Studio Visualizer 2024. Prior to calculation, the protein backbone was aligned to ensure proper superposition. RMSD values were computed based on heavy atoms of the ligand, excluding hydrogens. The overlay of the re-docked and crystallographic ligands is shown in Fig. S8.” (lines 597–601)

Furthermore, the recalculated RMSD value of 0.62 Å has been incorporated into the Results and Discussion section. This value demonstrates the close agreement between the docked and crystallographic conformations of the ligand, thereby confirming the reliability and accuracy of the docking protocol used in this study.

The added text reads as follows: To validate the reliability of our docking protocol, we conducted a redocking study using the co-crystallized ligand, lactose-3′-sulfate, applying the same docking parameters used for our test compounds. The ligand was redocked into the binding site, and the resulting pose was compared with the crystallographic conformation. The re-docking closely reproduced the experimental pose with a root-mean-square deviation (RMSD) of 0.62 Å (Fig. S8) and a docking energy of -7.0 kcal/mol. The low RMSD value demonstrates a close agreement between the docked and crystallographic poses, confirming that our docking procedure, including the chosen parameters, grid definition, and preparation protocols, accurately captures the key interactions and binding orientation of the ligand. (lines 311–319)